# Photocrosslinking-induced CRAC channel-like Orai1 activation independent of STIM1

Lena Maltan[1,3], Sarah Weiß[1,3], Hadil Najjar[1,3], Melanie Leopold [1],
Sonja Lindinger[1], Carmen Höglinger[1], Lorenz Höbarth[1], Matthias Sallinger [1],
Herwig Grabmayr [1], Sascha Berlansky[1], Denis Krivic[2], Valentina Hopl[1],
Anna Blaimschein[1], Marc Fahrner [1], Irene Frischauf[1], Adéla Tiffner[1] &
Isabella Derler [1] ✉

Ca²⁺ release-activated Ca²⁺ (CRAC) channels, indispensable for the immune system and various other human body functions, consist of two transmembrane (TM) proteins, the Ca²⁺-sensor STIM1 in the ER membrane and the Ca²⁺ ion channel Orai1 in the plasma membrane. Here we employ genetic code expansion in mammalian cell lines to incorporate the photocrosslinking unnatural amino acids (UAA), p-benzoyl-L-phenylalanine (Bpa) and p-azido-L-phenylalanine (Azi), into the Orai1 TM domains at different sites. Characterization of the respective UAA-containing Orai1 mutants using Ca²⁺ imaging and electrophysiology reveal that exposure to UV light triggers a range of effects depending on the UAA and its site of incorporation. In particular, photoactivation at A137 using Bpa in Orai1 activates Ca²⁺ currents that best match the biophysical properties of CRAC channels and are capable of triggering downstream signaling pathways such as nuclear factor of activated T-cells (NFAT) translocation into the nucleus without the need for the physiological activator STIM1.

Calcium ions (Ca²⁺) are crucial messengers in a variety of signaling pathways to control a wide range of cellular events such as proliferation, gene expression and muscle contraction. In addition, they play an essential role in development of various pathophysiological processes, emphasizing the importance of precisely determining the dynamics of specific cellular Ca²⁺ signaling cascades of different Ca²⁺ transport mechanisms[1].

An important Ca²⁺ entry pathway, the so-called Ca²⁺ release-activated Ca²⁺ (CRAC) channel[2,3] is formed by its major constituents, the Stromal Interaction Molecule (STIM1, 2) and the Orai protein (Orai1, 2, and 3). STIM, a single-pass membrane protein anchored in the endoplasmic reticulum (ER) membrane, triggers the Ca²⁺ influx through the highly selective Orai channel located in the plasma membrane[4–6]. Published structures of Orai channels have consistently revealed their unique hexameric assembly[7–10], in which each subunit

consists of four transmembrane (TM) domains, one intracellular and two extracellular loop regions, and a cytosolic N- and C- terminus. Orai activation is initiated by receptor-induced ER depletion, sensed by the luminal domain of STIM, which results in oligomerization and translocation of STIM to the ER-PM junctions. It directly couples to Orai C-termini[2,11,12] to initiate pore opening and Ca²⁺ influx[13,14].

Of particular interest in CRAC channel research is the dissection of the distinctive TM domain motions within and between adjacent Orai subunits that account for the typical biophysical properties of the channel[15–18]. Structural resolutions of the Drosophila melanogaster Orai channel, which is 73% homologous to human Orai1, showed that pore opening is associated not only with conformational changes along the pore but also with rigid body movements of each subunit[7]. Our previous investigations revealed that a global conformational change of all Orai TM domains is indispensable for Orai1 pore opening,

[1]Institute of Biophysics, JKU Life Science Center, Johannes Kepler University Linz, A-4020 Linz, Austria. [2]Division of Medical Physics and Biophysics, Gottfried Schatz Research Center, Medical University of Graz, A-8010 Graz, Austria. [3]These authors contributed equally: Lena Maltan, Sarah Weiß, Hadil Najjar.
✉e-mail: Isabella.derler@jku.at

due to a dominant effect of several loss-of-function (LoF) over a set of gain-of-function (GoF) point mutations[14]. This indicates that a variety of critical checkpoints and intrinsic inter- and intramolecular TM domain interactions are required for the establishment of an opening-permissive channel architecture[14]. However, the functional and structural studies available in this regard provide only static snapshots and information on the opening dynamics and the functional role of transient interactions is still rare.

A real breakthrough was previously achieved by integrating optogenetic tools to the key players of the CRAC channel. Remarkably, linking light-switchable proteins to STIM1 or Orai1 enabled remote and reversible stimulation of their activation by light independently of store-depletion. This approach has provided precise control over CRAC channel downstream signaling and significantly advanced the understanding of the STIM1 activation machinery[19–22]. Moreover, optopharmacological approaches opened new avenues in deciphering the molecular basis of physiological processes and recently enabled precise manipulation of the CRAC channel. Azobenzene-containing photoswitches were designed to specifically switch CRAC channels and downstream signaling on and off[23,24].

The use of the emerging optoproteomics technology is an alternative strategy to transfer light-sensitivity not only to a protein of interest but to individual amino acids[25]. This involves the site-specific incorporation of light-sensitive unnatural amino acids (UAAs) by the genetic code expansion (GCE) technology[25–27]. UAA incorporation relies on the re-assignment of a nonsense codon (traditionally the amber stop codon (TAG)) in the gene of interest by a bio-orthogonal engineered suppressor tRNA which is aminoacylated with the designated UAA. These tRNA/tRNA synthetase (aaRS) pairs allow direct and efficient insertion of genetically-encoded UAAs into the protein of interest[25–27]. This approach provides real-time analysis of the protein's structure-function relationship at the amino acid level, which to date has been applied to only a handful of membrane proteins[27–36].

In this study, we utilize photocrosslinking UAAs to achieve temporally precise, light-mediated remote control over Orai1 channel activation at the level of single amino acids. Compared to chemical crosslinking, photocrosslinking UAAs have the advantage of being applicable not only to solvent-accessible sites, but also to membrane-embedded channel domains[37,38]. We demonstrate that the insertion of photocrosslinking UAAs in Orai1 at various positions within the TM domains is a flexible and efficient method to confer light-sensitivity to the entire channel complex. In particular, our approach enables CRAC channel-like activation independent of STIM1 via photocrosslinking at different sites in the TM domains capable of triggering $Ca^{2+}$-dependent downstream signaling events.

## Results

### Insertion of photocrosslinking UAAs in TM2, TM3 and TM4 enables Orai1 activation upon UV light irradiation

Orai1 pore opening requires a series of checkpoints in all TM domains to adopt an opening-permissive conformation, as evidenced by a number of known GoF- and LoF-Orai1 mutations[14,39–46]. We hypothesized that these positions in particular would be sufficiently flexible to confer light-sensitivity to Orai1 via inserting photocrosslinking UAAs. These could trigger local conformational changes through photocrosslinking, which would then be transferred to the entire channel complex to eventually cause pore opening. We chose the photocrosslinking UAAs, p-azido-L-phenylalanine (Azi) and p-benzoyl-L-phenylalanine (Bpa) (Fig. 1a), which become reactive upon irradiation with UV light (365 nm) and form covalent bonds (e.g. C-H) with residues in the immediate vicinity of 3-4Å[37,38].

By incorporating these photocrosslinking UAAs (Azi, Bpa), we primarily screened positions in Orai1 TM domains (Fig. 1b), already known as critical checkpoints[14,39], for UV-induced changes in cellular $Ca^{2+}$ concentrations. To ensure that the Orai1 mutants were not

truncated at the TAG stop codon, the fluorescence label was C-terminally tethered. Figure 1 and Supplementary Fig. 1 show the basic outcome of our $Ca^{2+}$ fluorescence measurements for 20 UAA insertion sites, all exhibiting maintained plasma membrane expression comparable to wild-type (Supplementary Fig. 2).

6 of the 20 positions containing Azi and 7 of the 20 positions containing Bpa led to constitutive activation prior to UV light irradiation, as monitored by R-GECO1.2. 7 mutants containing Azi and 7 mutants containing Bpa exhibited enhanced $Ca^{2+}$ levels upon exposure to UV light - among those 4 with Azi and 4 with Bpa significantly. Interestingly, 4 positions containing Azi and 4 positions containing Bpa triggered constitutive activity in the non-excited state, which was partially reduced by UV light irradiation. All other Orai1 mutants containing UAA substitutions at distinct positions showed no significant $Ca^{2+}$ influx neither before nor after UV light irradiation like Orai1 wild-type (Fig. 1, Supplementary Fig. 1).

Here, we focused on three light-sensitive Orai1 mutants, specifically Orai1 A137Bpa (TM2), Orai1 L174Bpa (TM3) and Orai1 A254Azi (TM4), as they showed robust UV light-induced activation. Each mutant contained the photocrosslinking UAA in a distinct TM domain surrounding the pore-lining TM1. $Ca^{2+}$ imaging experiments showed that $Ca^{2+}$ levels increased upon the exchange from 0 mM to 2 mM $Ca^{2+}$-containing extracellular solution in Orai1 A137Bpa and Orai1 A254Azi, but not Orai1 L174Bpa overexpressing cells. Subsequent application of a 10 s UV pulse significantly enhanced $Ca^{2+}$ entry, while a second 30 s UV light pulse did not further increase cytosolic $Ca^{2+}$ concentrations (Fig. 2a–c; Supplementary Fig. 3a–d).

Complementary electrophysiological recordings revealed no or weak constitutive $Ca^{2+}$ currents of the three light-sensitive Orai1 mutants. In agreement with $Ca^{2+}$ imaging studies, all showed an instantaneous increase to maximum values upon UV light irradiation for 15 s, which did not occur for wild-type Orai1 (Fig. 2d–f, Supplementary Fig. 3e, f). Moreover, all three mutants exhibited selective $Ca^{2+}$ currents and a reversal potential ($V_{rev}$) of around +50 mV, similar to wild-type STIM1/Orai1 currents (Fig. 2g–i). Application of increasing UV intensities led to comparable maximum activation (Supplementary Fig. 3d, i–k), but with a significant reduction in activation time after switching UV light on (Supplementary Fig. 3i–k). Regarding strongly reduced constitutive activity of these mutants in patch-clamp compared to $Ca^{2+}$ imaging experiments, it is notable that only highly transfected cells exhibited constitutive activity which enhanced drastically after exposure to UV light. However, such cells were only rarely detected (~5%) (Supplementary Fig. 4a, b). Interestingly, Orai1 A254Azi exhibited slow inactivation upon maximum UV-mediated activation, the extent of which depends on the level of $Ca^{2+}$ current density. Indeed, increasing extracellular $Ca^{2+}$ levels (Supplementary Fig. 4c–e) or UV intensities enhanced $Ca^{2+}$ currents and increased the extent of slow inactivation of UV-mediated Orai1 A254Azi currents (Supplementary Fig. 4f–g).

In control experiments, we ensured that an intact plasma membrane expression of the mutants could only be achieved in the presence of the UAA and the expression of the respective tRNA/aaRS pair. In the absence of the latter, only weak background fluorescence was detectable in the cytosol, but not in the plasma membrane (Supplementary Fig. 5a) in line with abolished UV-mediated $Ca^{2+}$ influx comparable to untransfected cells (Supplementary Fig. 5b–e). Furthermore, examination of the UV-evoked $Ca^{2+}$ currents of cells co-expressing Orai1 A137Bpa-YFP and Orai1-CFP wild-type as a function of the Orai1 A137Bpa:Orai1 fluorescence intensity ratio showed a tendency for higher maximum currents with increasing ratio and highest UV-induced activation when only Orai1 A137Bpa was expressed (Supplementary Fig. 6a), indicating efficient UAA incorporation. Moreover, rapid UV-mediated activation occured independent of endogenous STIM and Orai isoforms, as shown in CRISPR/Cas9 STIM1/STIM2 double-knockout (DKO), CRISPR/Cas9 STIM1/Orai1 DKO and CRISPR/Cas9 Orai1/Orai2/Orai3 triple KO (TKO) cells (Supplementary Fig. 6c, d),

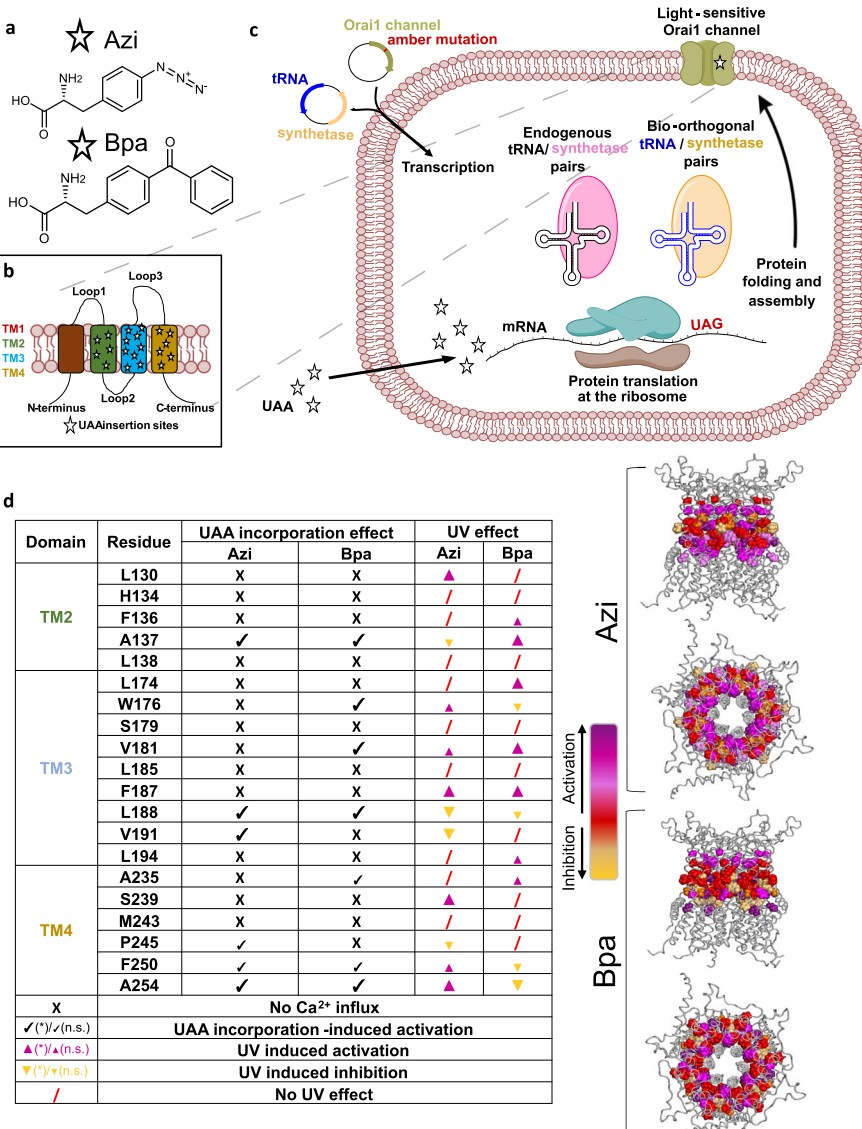

**Fig. 1 | Site-specific incorporation of Azi and Bpa in Orai1. a** Chemical structures of Azi and Bpa. **b** Scheme of the TM domains of an Orai1 subunit. Stars indicate sites for UAA incorporation. **c** Principle of UAA incorporation in mammalian cells. A gene encoding a protein of interest (here Orai1) containing an Amber stop codon (TAG) inserted at the desired site is co-transfected with a bioorthogonal tRNA (blue)/aminoacyl synthetase (RS) pair (yellow) that does not crosstalk with endogenous pairs (black/pink). Cells are incubated with the UAA supplemented in the medium. Within the host cell, the suppressor tRNA, aminoacylated with the UAA by the bioorthogonal RS, recognizes the Amber stop codon on the mRNA (UAG codon, red) at the ribosome to insert the UAA into the nascent amino acid chain. Different light-sensitive Orai1 mutants are generated carrying the photocrosslinking UAAs at the desired sites. **d** The table summarizes the effects on Ca²⁺ influx of all screened Orai1 mutants containing Azi or Bpa at a position in TM2-4 after UAA incorporation and after UV light (365 nm) irradiation. A cross indicates no activity (below threshold line) after UAA insertion. Small (no significant change in activity (n.s.), but above threshold line (see Supplementary Fig. 1)) or large (significant change in activity and above threshold line (*)) check marks indicate constitutive activity upon switching from a 0 mM to a 2 mM Ca²⁺-containing solution. UV illumination (10 s) can lead to activation which is highlighted by small (no significant change in activity before versus after UV light, but above threshold line (n.s.)) or large (significant change in activity before after after UV light and above threshold (*)), purple upward-facing triangles. UV-induced deactivation is illustrated by small (reduced activity, but not significant (n.s.)) or large (significantly reduced activity (*)) yellow downward-facing triangles. Mutants with no change in activity upon UV illumination are marked with a red slash. Accordingly, UV-induced effects (indicated by the color gradient) are represented in the top- and side-view cartoons of Orai1 in the respective colors. Corresponding data are shown in Supplementary Fig. 1 and provided as a Source Data file.

although to slightly lower maximum current levels, likely owing to lower expression levels in the KO compared to wild-type HEK293 cells (Supplementary Fig. 6e). Moreover, we showed that ER Ca²⁺ levels detected by LAR-GECO1 in HEK293 cells expressing either Orai1 or Orai1 A137Bpa compared to untransfected cells remained comparable before and after UV light irradiation and subsequent thapsigargin (TG)-induced store-depletion led to a similar decrease in Ca²⁺ levels (Supplementary Fig. 6f), thus excluding that UV light-triggered Ca²⁺ entry leads to Ca²⁺ overload in the ER.

We conclude that insertion of photocrosslinking UAAs at specific positions into Orai1 TM domains and subsequent activation by UV light is suitable to trigger the activation of inwardly rectifying Ca²⁺ currents similar to wild-type STIM1/Orai1 currents.

### Photocrosslinking-induced activation occurs independently of STIM1

Consecutively, we investigated whether the photocrosslinking UAA-containing Orai1 variants are affected by STIM1.

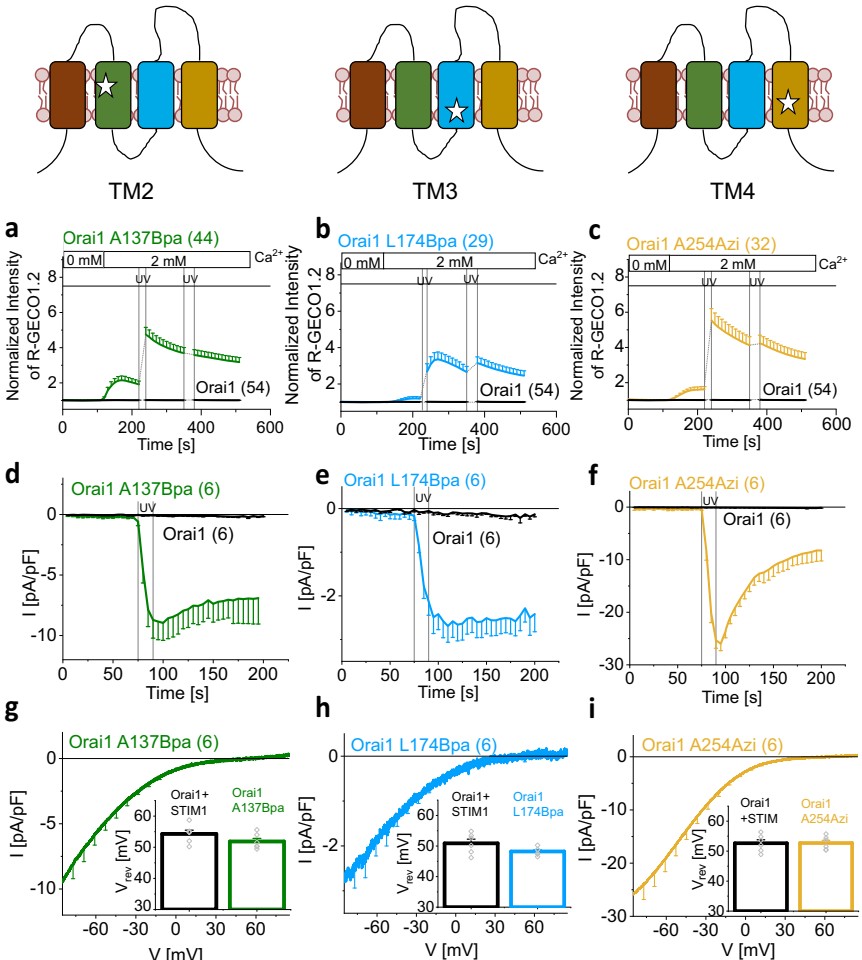

**Fig. 2 | Photocrosslinking of Orai1 TM domains can lead to CRAC channel-like activation.** Schemes show one Orai1 subunit representing either Orai1 A137Bpa, Orai1 L174Bpa or Orai1 A254Azi in the absence of STIM1. Stars indicate sites for UAA incorporation. $Ca^{2+}$ imaging measurements of Orai1 A137Bpa (**a**), Orai1 L174Bpa (**b**) and Orai1 A254Azi (**c**) using R-GECO1.2. Intracellular $Ca^{2+}$ levels, represented by the normalized intensity of R-GECO1.2 co-transfected with the above mentioned UAA-containing Orai1 mutants in HEK 293 cells, were monitored initially in 0 mM $Ca^{2+}$ solution followed by a 2 mM $Ca^{2+}$ solution. Under 2 mM $Ca^{2+}$ solution conditions, UV light was applied for 10 s and 30 s, respectively. **d**–**f** Time courses of $Ca^{2+}$ current-densities after whole-cell break-in of above mentioned UAA-containing Orai1 mutants. UV light was applied for 15 s during the recording leading to UV-induced activation of $Ca^{2+}$ currents. **g**–**i** Corresponding current/voltage (I/V) relationships were taken after 100 s of measurement. Inlet represents reversal potential ($V_{rev}$) of wild-type CRAC channel (Orai1 + STIM1) versus light-sensitive Orai1 mutant currents. Single values are indicated in gray. Data represent mean values ± SEM of indicated number (n) of experiments. Mann–Whitney test was applied to show that $V_{rev}$ are not significantly different. Detailed statistic values are shown in Supplementary Table 3. Source data are provided as a Source Data file.

All three mutants allowed STIM-mediated activation, similar to wild-type Orai1 (Fig. 3a–c). Subsequent exposure to UV light resulted in further current enhancements (Fig. 3a–c), which did not occur for STIM1-activated wild-type Orai1, both in normal and STIM1/Orai1 DKO HEK293 cells (Fig. 3a–c, Supplementary Fig. 3g, h, Supplementary Fig. 6g, h). $Ca^{2+}$ currents before and after UV light activation exhibited robust inward rectification and a $V_{rev}$ in the range of +50 mV, comparable to Orai1 mutants in the absence of STIM1 and wild-type currents (Fig. 3d–f). Conversely, we first exposed cells co-expressing STIM1 and one of the three photocrosslinking UAA-containing Orai1 mutants to UV light prior to passive store-depletion. Under these conditions, Orai1 A137Bpa showed maximal $Ca^{2+}$ current levels that did not increase further during passive store-depletion and reached higher levels than those obtained for STIM1-induced activation before and after UV light exposure (Fig. 3g, Supplementary Fig. 6g, h). In contrast, UV-activated currents of Orai1 L174Bpa and Orai1 A254Azi further increased during subsequent passive store-depletion, reaching comparable maximum values when the sequence of STIM1-mediated activation and UV light exposure was reversed (Fig. 3h, i). STIM1 coupling upon store-depletion to a photocrosslinking UAA-containing mutant,

occured to comparable levels as for wild-type Orai1 without as well as with exposure to UV light, as exemplarily shown for Orai1 A137Bpa by FRET microscopy (Fig. 3j–m). Corresponding $Ca^{2+}$ imaging experiments exemplified by Orai1 A137Bpa exhibited store-operated activation in the presence of STIM1, which slightly enhanced upon subsequent exposure to UV light (Supplementary Fig. 7a). Vice versa, initial exposure to UV light led to full activation of Orai1 A137Bpa, which did not further enhance upon ensuing passive store-depletion (Supplementary Fig. 7b). In contrast to our electrophysiological results (Fig. 3g), $Ca^{2+}$ levels obtained for Orai1 A137Bpa revealed no clear difference between the STIM1 unbound and bound state (Supplementary Fig. 7a, c). We hypothesized that this observation was likely due to the full saturation of R-GECO1.2 with $Ca^{2+}$ under activated conditions. Indeed, when 0.5 mM $Ca^{2+}$ was used instead of 2 mM $Ca^{2+}$ in the extracellular solution, a significantly lower $Ca^{2+}$ level of Orai1 A137Bpa was observed upon activation by STIM1 compared to the absence of STIM1 and subsequent exposure to UV light (Supplementary Fig. 7c, d). Overall, UAA insertion did not interfere with STIM1-mediated activation and STIM1 coupling could modulate the activity of photocrosslinking UAA-containing Orai1 mutants. Noteworthy, in particular

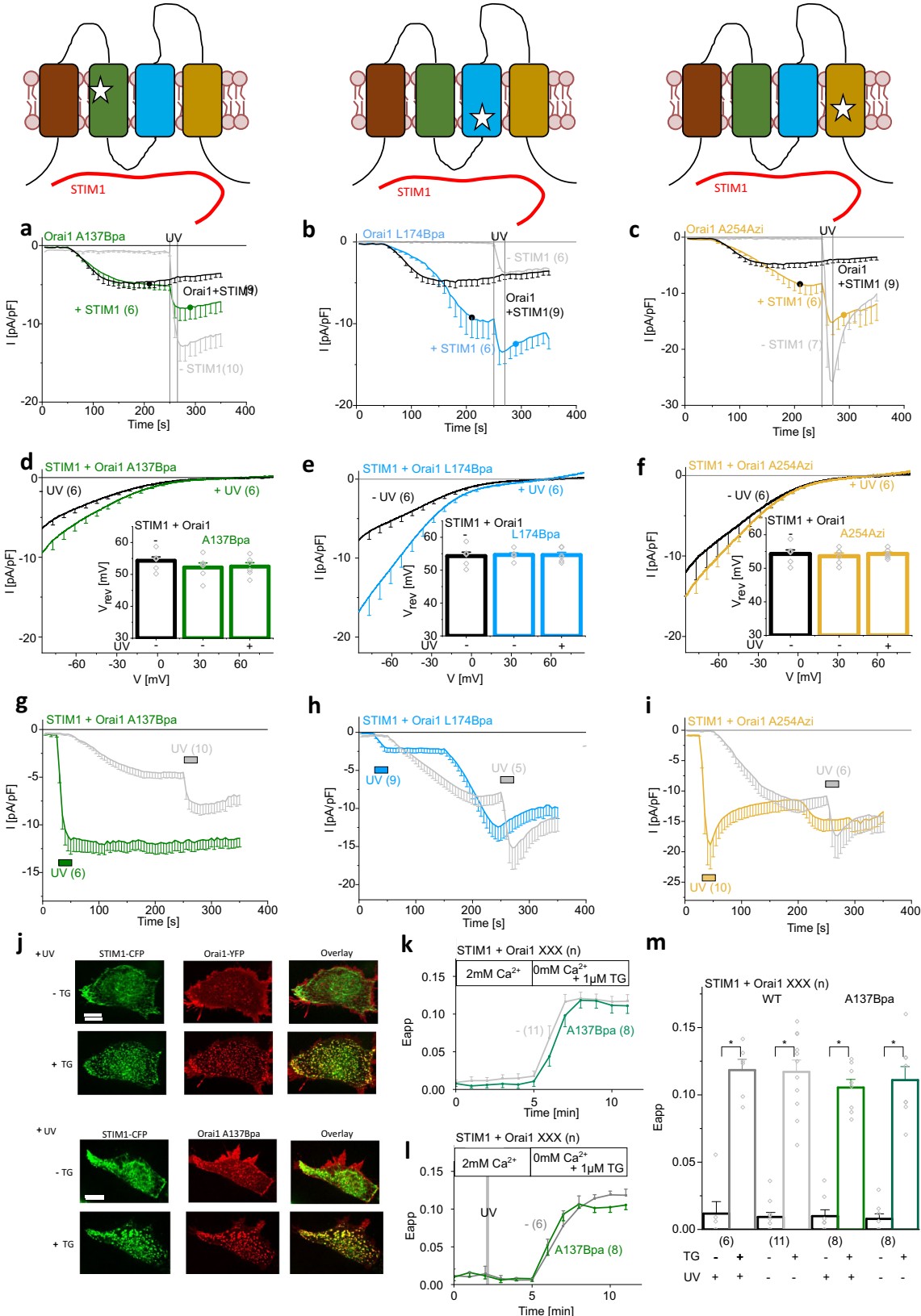

for Orai1 A137Bpa and Orai1 A254Azi, UV light irradiation seems to be sufficient for maximal Ca²⁺ current activation.

In accordance with the latter, mutation of the major STIM1-binding site within the C-terminus (L273D[47]) of Orai1 A137Bpa, Orai1 L174Bpa, and Orai1 A254Azi resulted in comparable UV-mediated activation, both in the absence and the presence of STIM1 (Fig. 4a, b;

Supplementary Fig. 7e). In analogy, also the co-expression of STIM1 mutants (STIM1 L373S ± A376S[48]), which are deficient in coupling to Orai1, allowed UV-mediated activation of Orai1 A137Bpa or Orai1 A137Bpa L273D to a comparable extent as in the absence of these STIM1 mutants (Fig. 4a, b; Supplementary Fig. 7f, g). Consistent with these electrophysiological studies, Ca²⁺ levels of cells expressing Orai1

**Fig. 3 | STIM1 modulates the activity of photocrosslinking UAA-containing Orai1 mutants.** Schemes show one Orai1 subunit representing either Orai1 A137Bpa, Orai1 L174Bpa or Orai1 A254Azi in the presence of STIM1. Stars indicate sites for UAA incorporation. **a**–**c** Time courses of Ca$^{2+}$ current densities after whole-cell break-in of the above mentioned Orai1 mutants co-expressed with STIM1. UV light was applied for 15 s leading to UV-induced Orai1 activation after reaching maximum STIM1-mediated Orai1 mutant activation. Time courses in light gray show the respective Orai1 mutants in the absence of STIM1. **d**–**f** Corresponding I/V relationships were taken from indicated time points (black and colored circles, respectively) in (**a**–**c**). Inlet represents the reversal potential (V$_{rev}$) of STIM1-activated Orai1 channel currents versus STIM1-activated light-sensitive Orai1 mutants currents before and after application of UV light. Single values are indicated in gray. **g**–**i** Time courses of current densities after whole-cell break-in of the above mentioned UAA-containing Orai1 mutants co-expressed with STIM1. UV light was applied for 15 s at time segments indicated by colored bars, either before or after passive store-depletion mediated activation. **j** Confocal fluorescence microscopy images of representative cells before and after treatment with 1 μM thapsigargin (TG) showing STIM1-CFP, Orai1-YFP or Orai1 A137Bpa-YFP as well as their overlay. Images were recorded after the application of 10 s UV light. White bars indicate 5 μm. Time courses of FRET (E$_{app}$) monitoring the interaction of STIM1 with wild-type Orai1 or Orai1 A137Bpa when switching from a 2 mM Ca$^{2+}$-containing solution to a 0 mM Ca$^{2+}$/1μM TG solution inducing STIM1/Orai1 interaction without (**k**) or after (**l**) application of 10 s UV light. **m** Corresponding bar diagram to (**k**) and (**l**) comparing FRET of STIM1/wild-type Orai1 and STIM1/Orai1 A137Bpa before and after treatment with 1 μM TG as well as with and without application of UV light (Welch-ANOVA for (m) + Fig. 4c: F(7;20,37)=86,17, p = 1.53*10$^{-13}$). Data represent mean values ± SEM of indicated number (n) of experiments. *Significant differences (p < 0.05). Detailed statistic values are shown in Supplementary Table 3. Source data are provided as a Source Data file.

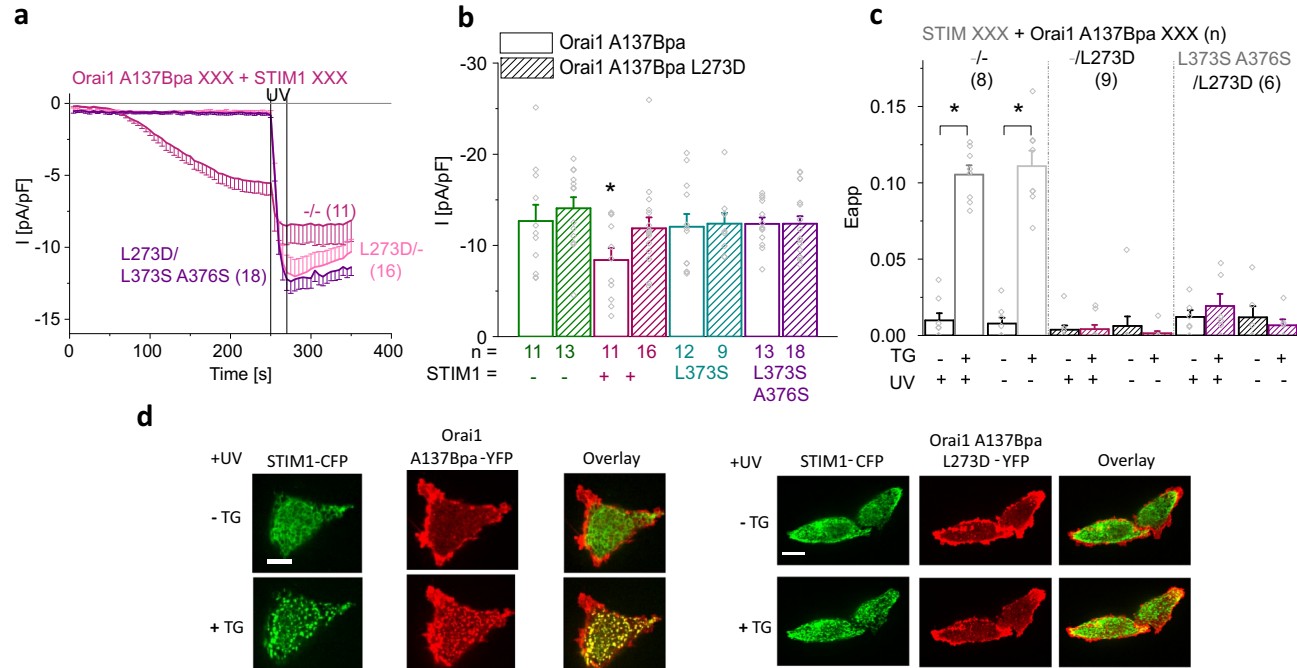

**Fig. 4 | STIM1 is not required for UV-mediated activation of photocrosslinking UAA-containing Orai1 mutants. a** Time courses of current densities after whole-cell break-in of Orai1 A137Bpa or Orai1 A137Bpa L273D co-expressed with either STIM1 or STIM1 L373S A376S. UV light is applied for 15 s at t = 250 s, the time point after maximum STIM1-mediated Orai1 mutant activation indicating completed store-depletion. **b** Bar diagram summarizes UV-induced maximum currents measured for Orai1 A137Bpa or Orai1 A137Bpa L273D in the absence of STIM1 or the presence of STIM1, STIM1 L373S or STIM1 L373S A376S. Single values are indicated in gray (one-way ANOVA for (**b**): F(7;99) = 2,15, p = 0.046). **c** Bar diagram of FRET (E$_{app}$) comparing STIM1 or STIM1 L373S A376S co-expressed with Orai1 A137Bpa or Orai1 A137Bpa L273D before and after treatment with 1 μM TG as well as with and without application of UV light (Welch-ANOVA for (**c**) + Fig. 3m: F(7;20,37) = 86.17, p = 1.53*10$^{-13}$). **d** Confocal fluorescence microscopy images of representative cells before and after treatment with 1 μM TG showing STIM1-CFP, Orai1 A137Bpa-YFP or Orai1 A137Bpa L273D-YFP as well as an overlay of both after application of 10 s UV light. White bars indicate 5 μm. Data represent mean values ± SEM of indicated number (n) of experiments. *Significant differences (p < 0.05). Detailed statistic values are shown in Supplementary Table 3. Source data are provided as a Source Data file.

A137Bpa L273D and STIM1 L373S A376S were significantly enhanced upon UV light irradiation (Supplementary Fig. 7d). Remarkably, activation by UV light led under all conditions with defective STIM1/Orai1 A137Bpa binding to significantly higher levels than in the presence of intact STIM1 coupling (Fig. 4a, b; Supplementary Fig. 7d; Supplementary Fig. 7f,g), notably in Ca$^{2+}$ imaging experiments in the presence of 0.5 mM Ca$^{2+}$, but not 2 mM Ca$^{2+}$ in the extracellular solution (Supplementary Fig. 7c, d). Moreover, varying the ratio (Orai1 A137Bpa:STIM1) of the fluorescence intensity of co-expressed STIM1-CFP and Orai1 A137Bpa-YFP revealed that the higher the amount of STIM1, the smaller the UV-mediated increases in the currents after STIM1-mediated activation (Supplementary Fig. 6b). In control experiments, we confirmed that above mentioned STIM1 and Orai1 C-terminal point mutants

impair coupling to Orai1 A137Bpa, as exemplified by STIM1 or STIM1 L373S A376S co-expressed with Orai1 A137Bpa L273D (Fig. 4c, d). Taken together, this indicates that deficient STIM1/Orai1 binding does not interfere with photocrosslinking-induced activation. Interestingly, STIM1-mediated activation followed by exposure to UV-light resulted in reduced maximum activation levels of Orai1 A137Bpa compared to UV activation in the STIM1-unbound state.

Furthermore, we investigated whether refilling of the ER stores, known to shut down Orai1 activation due to disassembly of the STIM1/Orai1 complex[12,49], affects UV light-activated Orai1 mutant activation using Ca$^{2+}$ imaging. Store-depletion by 2,5-di-t-butylhydroquinone (BHQ)/carbachol (CCH) triggers store-operated Ca$^{2+}$ level enhancements of Orai1 A137Bpa similar to wild-type Orai1 in the presence of

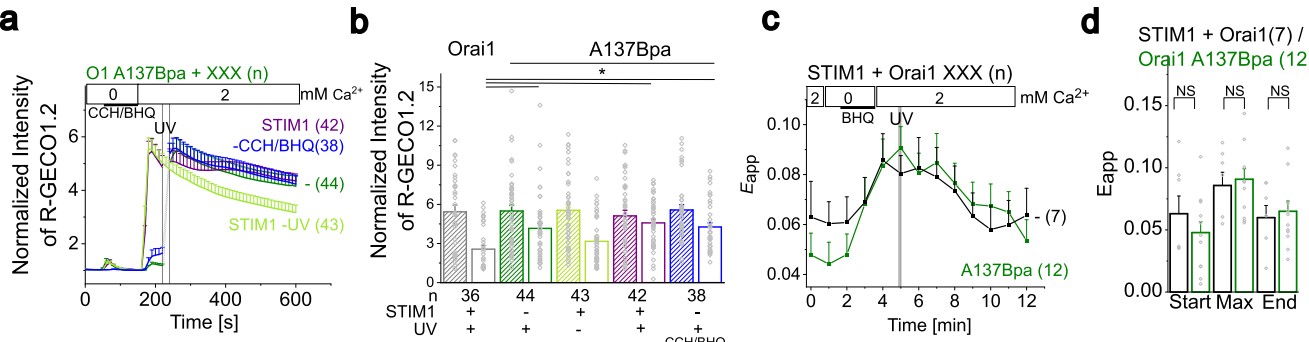

**Fig. 5 | Photocrosslinking induced Orai1 activation delays the return to the resting state and leaves STIM1 coupling unaffected. a** Ca²⁺ imaging measurements showing intracellular Ca²⁺ levels, represented by the normalized intensity of R-GECO1.2 co-transfected with Orai1 A137Bpa either without or with STIM1, while switching from a 0 mM Ca²⁺ to a 0 mM Ca²⁺/100 µM CCH/50 µM BHQ and finally to a 2 mM Ca²⁺-containing solution. Blue trace shows Orai1 A137Bpa currents without 100 µM CCH/50 µM BHQ application. UV light was applied for 10 s to cells expressing Orai1 A137Bpa or STIM1 + Orai1 A137Bpa and compared to STIM1 + Orai1 A137Bpa evoked Ca²⁺ levels in the absence of UV light. **b** Summarizing bar diagram comparing Ca²⁺ levels in (**a**) and of STIM1 + wild-type Orai1 at maximal levels and at

t = 600 s (one-way ANOVA for (**b**): $F(9;160,73) = 9,79$, $p = 6,96*10^{-12}$). **c** Time courses of FRET ($E_{app}$) values monitoring the interaction of STIM1 with wild-type Orai1 or Orai1 A137Bpa when switching from a 2 mM Ca²⁺ solution to a 0 mM Ca²⁺/10 µM BHQ solution and subsequently to a 2 mM Ca²⁺ solution. UV light irradiation was applied for 10 s subsequent to store-depletion in 2 mM Ca²⁺ solution. **d** Summarizing bar diagram of FRET ($E_{app}$) values corresponding to (**c**) at indicated time points (t = 0 (Start), 6 (Max), 12 min (End)). Data represent mean values ± SEM of indicated number (*n*) of experiments. *Significant differences ($p < 0.05$). Detailed statistic values are shown in detail in Supplementary Table 3. Source data are provided as a Source Data file.

STIM1. Subsequent washout of BHQ/CCH and thus refilling of ER stores showed only without exposure to UV light a comparable decay of maximum Ca²⁺ levels as for wild-type STIM1/Orai1 (Fig. 5a, b). UV-triggered Ca²⁺ level enhancements of Orai1 A137Bpa remained at significantly higher levels over time, both in the absence (without/with application of BHQ/CCH) and presence of STIM1. This is in line with findings mentioned above that photocrosslinking-induced Orai1 activation is independent of STIM1 and the return of STIM1 to the resting conformation (Fig. 5a, b).

Next, we investigated the effect of ER store refilling on STIM1 coupling to UV-activated Orai1 A137Bpa. In analogy to store-depletion by TG, also BHQ triggers STIM1 coupling to Orai1 A137Bpa to a comparable extent as for wild-type Orai1, which was not significantly enhanced after UV light irradiation. Upon washout of BHQ, FRET values for STIM1 and Orai1 A137Bpa decreased to a comparable extent as observed for wild-type STIM1 and Orai1 (Fig. 5c, d).

Overall, we conclude that photocrosslinking-induced Orai1 activation occurs independently of STIM1 and does not support STIM1 coupling. Nevertheless, initial STIM1 binding to a photocrosslinking UAA-containing Orai1 mutant is able to modulate UV light-induced activation.

## GoF-TM mutations, but not GoF-nexus mutations, impair photocrosslinking-induced Orai1 activation

As an alternative to STIM1-mediated Orai1 activation, we investigated the effect of GoF mutations leading to constitutively open Orai1 channels on photocrosslinking-induced activation. We focused particularly on known mutations in different Orai1 TM domains, that lead to strong constitutive currents in the absence of STIM1. Specifically, in TM1 we chose the single point mutation V102A, which has been reported to lead to constitutive, non-selective Orai1 currents in the absence of STIM1, while in the presence of STIM1, Orai1 V102A currents become selective with a $V_{rev}$ comparable to wild-type STIM1/Orai1 currents[50,51]. Interestingly, insertion of V102A into Orai1 A137Bpa, Orai1 L174Bpa and Orai1 A254Azi led to constitutive activity but did not allow further activation upon UV light irradiation, neither in the absence nor in the presence of STIM1 (Fig. 6a, b, Supplementary Fig. 8a). All double mutants exhibited a significantly higher $V_{rev}$ than Orai1 V102A even before UV light, which was close to the range of STIM1-Orai1 wild-type currents and did not change after UV activation and/or STIM1 coupling (Supplementary Fig. 8b–h). However, as the currents of the double

mutants did not increase upon UV irradiation (Fig. 6a, Supplementary Fig. 8a), it remains unclear whether photocrosslinking is disturbed potentially due to an altered conformation.

Next, we investigated the effect of other GoF single point mutations in the TM domains surrounding the pore-lining TM1, specifically, H134A (TM2)[41], V181K (TM3)[52] and P245L (TM4)[40]. They have been reported to lead to robust constitutive currents with enhanced Ca²⁺ selectivity compared to V102A[43]. Interestingly, most of these GoF point mutations negatively interfered with photocrosslinking-induced activation (Fig. 6a, b, Supplementary Fig. 8a). In particular, incorporation of any of these GoF mutations in Orai1 A137Bpa led to a drastically reduced function or LoF (Fig. 6a, b) even in the presence of STIM1 (Supplementary Fig. 8a). Investigation of a set of other reported GoF mutations (L130S[14], L138F[41], V181A, L185A and A235C[14,52]) revealed that only the insertion of V181A, known to trigger minor constitutive activity of Orai1 compared to e.g. H134A or V181K, or P245L in Orai1 A137Bpa (Orai1 A137Bpa V181A/P245L) still allowed at least partial UV-mediated activation (Supplementary Fig. 8i).

Insertion of H134A or V181K in Orai1 L174Bpa or Orai1 A254Azi led to constitutive activity, which, however, was not or only marginally enhanced by UV light irradiation (Fig. 6a). Similarly, these double mutants exhibited constitutive activity also in the presence of STIM1, which, however, was not further enhanced upon exposure to UV light (Supplementary Fig. 8a).

Remarkably, incorporation of P245L into Orai1 L174Bpa or Orai1 A245Azi (Orai1 L174Bpa P245L, Orai1 P245L A254Azi) led to constitutive activity, which was further enhanced upon UV light irradiation. In the presence of STIM1, constitutive activity of these double mutants enhanced upon passive store-depletion, but did not further increase upon UV light irradiation. This suggests that either UV light irradiation or STIM1 coupling are sufficient for pore opening (Fig. 6a, b, Supplementary Fig. 8a).

Furthermore, we investigated the prominent nexus mutation ₂₆₁ANSGA₂₆₅ (L261A V262N H264G K265A), located in the bent region connecting TM4 with the C-terminus at the periphery of the channel complex. It has been reported to trigger constitutive Orai1 currents with identical characteristics compared to wild-type STIM1/Orai1 currents[53]. Interestingly, incorporation of ANSGA in Orai1 A137Bpa, Orai1 L174Bpa and Orai1 A254Azi led to weak constitutive activity, which was further enhanced upon UV light irradiation to comparable extents like for the corresponding mutants in the absence of ANSGA. In

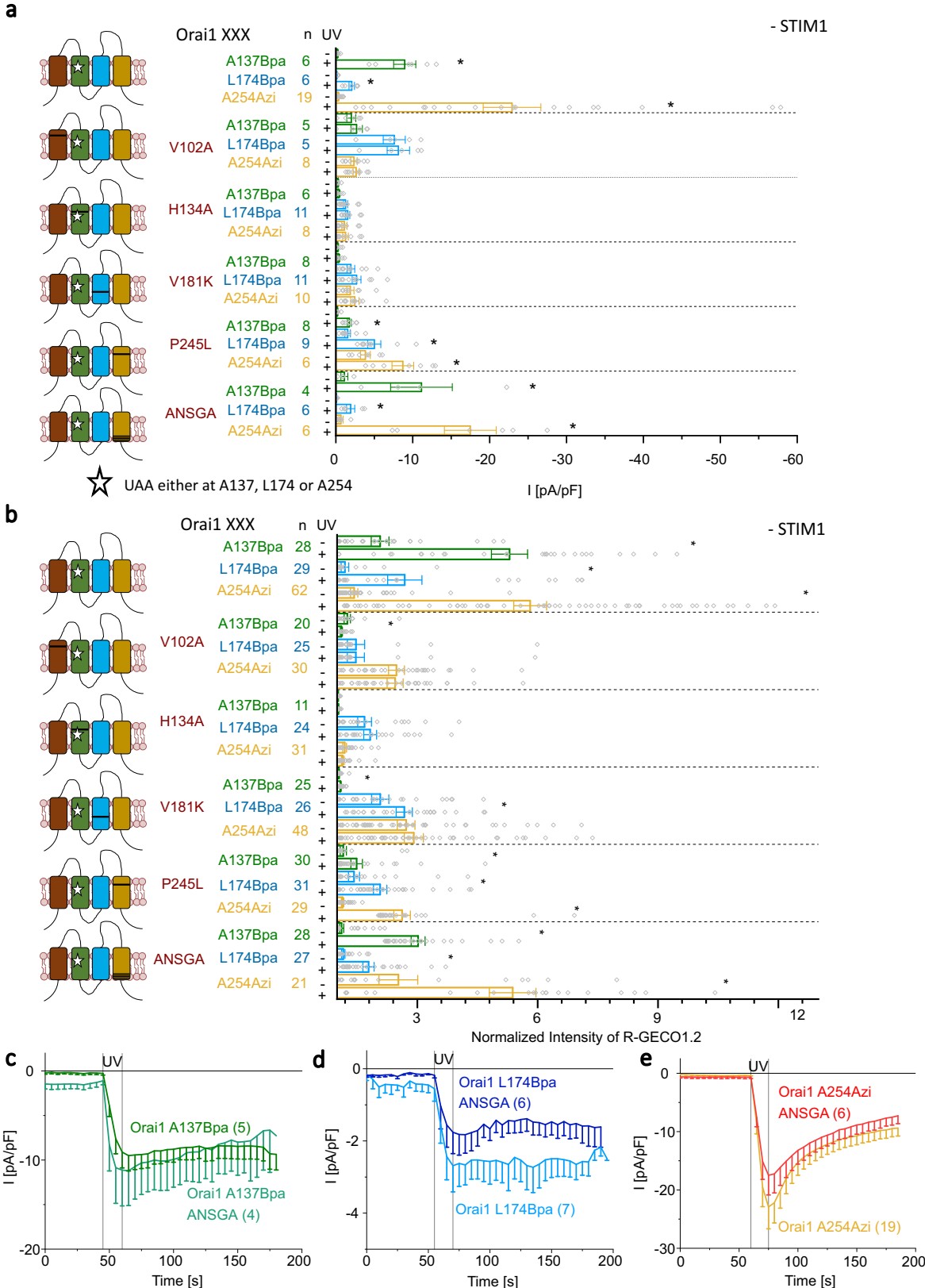

the presence of STIM1, all three mutants showed higher constitutive activity upon passive store-depletion which was further increased upon exposure to UV light (Fig. 6a–e, Supplementary Fig. 8a).

In summary, the incorporation of various Orai1 GoF mutations into the three light-sensitive Orai1 mutants containing a photo-crosslinking UAA in either TM2, TM3 or TM4, revealed that in particular the nexus mutation does not interfere with photocrosslinking-induced activation, neither in the absence nor presence of STIM1. All other GoF mutations in the different TM domains, except the slight constitutively active V181A and in part the P245L mutation, drastically interfered either with the general function or at least with photocrosslinking-induced activation.

**Fig. 6 | GoF mutations in Orai1 TM domains, but not Orai1 nexus mutation, interfere with UV-mediated activation of photocrosslinking UAA-containing Orai1 mutants.** Graphical illustration on the left-hand side represents photocrosslinking UAA-containing Orai1 mutants, exemplarily shown for Orai1 A137Bpa (star), combined with one of the following GoF mutations: V102A, H134A, V181K, P245L and $_{261}$ANSGA$_{265}$ (black line). **a** Bar diagram summarizes currents measured before and after application of UV light for the above mentioned photocrosslinking UAA- and GoF-containing Orai1 double mutants (Welch-ANOVA for Orai1 A137Bpa mutants: $F(11;21,41) = 7$, $p = 7.5*10^{-5}$; for Orai1 L174Bpa mutants $F(11;27,79) = 16,48$, $p = 1.96*10^{-9}$; for Orai1 A254Azi mutants $F(11;32,59) = 15,15$, $p = 6.72*10^{-10}$). **b** Bar

diagram summarizes normalized R-GECO1.2 intensities measured before and after application of UV light corresponding to (**a**) (Welch-ANOVA for (**b**): $F(35;326,09) = 27,11$, $p = 0$). Time course of current densities after whole-cell break-in of Orai1 A137Bpa $_{261}$ANSGA$_{265}$ compared to Orai1 A137Bpa (**c**), Orai1 L174Bpa $_{261}$ANSGA$_{265}$ compared to Orai1 L174Bpa (**d**) and Orai1 A254Azi $_{261}$ANSGA$_{265}$ compared to Orai1 A254Azi (**e**). UV light was applied for 15 s. Data represent mean values ± SEM of indicated number ($n$) of experiments. *Significant differences ($p < 0.05$). Detailed statistic values are shown in Supplementary Table 3. Source data are provided as a Source Data file.

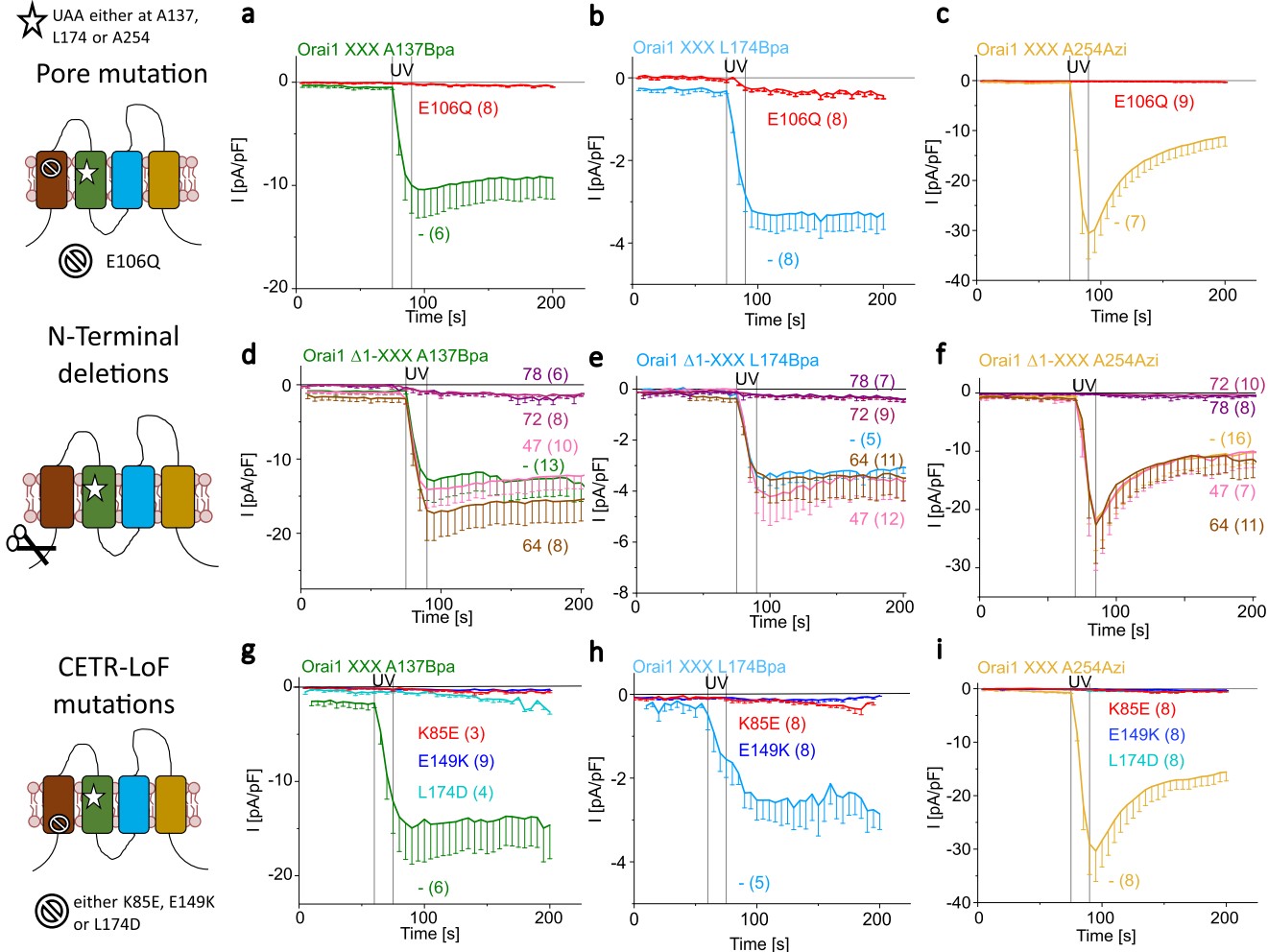

**Fig. 7 | Light-sensitive Orai1 mutants require an intact channel geometry.** Graphical illustration on the left-hand side represent photocrosslinking UAA-containing Orai1 mutants, exemplarily shown for Orai1 A137Bpa (star), combined with the pore mutation E106Q (stop sign), various N-terminal truncations (scissor) or cytosolic extended TM region (CETR)−LOF (stop sign) mutations. Time course of current densities after whole-cell break-in comparing Orai1 A137Bpa and Orai1 A137Bpa E106Q (**a**), Orai1 L174Bpa and Orai1 L174Bpa E106Q (**b**) and Orai1 A254Azi and Orai1 A254Azi E106Q (**c**). Time course of current densities after whole-cell break-in comparing the light-sensitive Orai1 mutants (Orai1 A137Bpa (**d**), Orai1

L174Bpa (**e**) and Orai1 A254Azi (**f**)) with different corresponding N-terminally truncated mutants (Orai1 A137Bpa Δ1-47/64/72/78; Orai1 L174Bpa Δ1-47/64/72/78; Orai1 A254Azi Δ1-47/64/72/78). Time course of current densities after whole-cell break-in comparing the light-sensitive Orai1 mutants (Orai1 A137Bpa (**g**), Orai1 L174Bpa (**h**) and Orai1 A254Azi (**i**)) with different corresponding mutants containing different CETR-LoF mutations (K85E, E149K and L174D). In all cases (a-i) UV light is applied for 15 s. Data represent mean values ± SEM of indicated number ($n$) of experiments. Source data are provided as a Source Data file.

## Photocrosslinking-induced activation requires an intact pore geometry and cytosolic regions

Apart from a series of gating checkpoints in Orai1 TM domains, an intact Orai1 pore opening is further determined by the selectivity filter[54], the N-terminus[12,43,51,55], cytosolic salt-bridge[14,53,56] and hydrophobic interactions[14,53]. In the following, we investigated whether these gating sites are also required for photocrosslinking-induced activation.

The selectivity filter of Orai1 is determined by E106 in TM1, whose mutation to E106Q leads to LoF[54]. Insertion of the point mutation E106Q into each light-sensitive Orai1 variant abolished UV light-mediated Ca$^{2+}$ current enhancements (Fig. 7a−c). This indicates the critical and dominant role of the selectivity filter also in photocrosslinking-induced Orai1 activation.

Intact Orai1 function necessitates at least the last 20 amino acids of the N-terminus (aa 70-90), otherwise Orai1 currents decrease

($\Delta N_{1-74}$) or vanish ($\Delta N_{1-76}$)[43,51]. We discovered that N-terminal deletion of the first 68 residues or less maintained photocrosslinking-induced activation of Orai1 A137Bpa, Orai1 L174Bpa and Orai1 A254Azi, at comparable or even higher levels compared to full-length channels. Increased truncations (Orai1 $\Delta N_{1-70/71}$ A137Bpa/L174Bpa/A254Azi) reduced the $Ca^{2+}$ influx after UV-mediated photocrosslinking while truncations after position 71 ($\Delta N_{1-72/78}$) completely abolished $Ca^{2+}$ currents (Fig. 7d–f; Supplementary Fig. 9a–e). Collectively, the conserved portion of the Orai1 N-terminus is not only indispensable for STIM1-mediated Orai1 activation[43,51], but also for photocrosslinking-induced Orai1 activation.

Finally, the cytosolic helical TM domain extensions include salt-bridge interactions between K85 and E149 and the hinge region (L174-L261), with hydrophobic L174 as critical determinants for pore opening. We showed that the LoF mutations K85E[57], E149K[14,56] and L174D[53] completely abolished UV-mediated currents of photocrosslinking UAA-containing mutants (Fig. 7g–i; Supplementary Fig. 9f, g) in line with previous findings[14,53,56,57].

Altogether, it is clear that photocrosslinking-induced Orai1 activation requires analogous key gating sites like STIM1-activated Orai1.

### Orai1 A137Bpa mimics authentic CRAC channel hallmarks best

To identify which of the three light-sensitive Orai1 mutants mimics STIM1-mediated Orai1 activation best, we compared the three most prominent biophysical CRAC channel characteristics. The latter include a $V_{rev}$ in the range of +50 mV, an enhancement of the currents upon switching from a $Ca^{2+}$-containing to a divalent-free (DVF) $Na^+$-containing solution and fast $Ca^{2+}$-dependent inactivation (FCDI)[43,58]. As shown in Fig. 2g–i, all three photocrosslinking UAA-containing mutants exhibited inward rectification with a comparable reversal potential to STIM1-activated Orai1 currents already in the absence of STIM1.

Typcially, STIM1/Orai1 currents show a 2-3-fold enhancement in the current density upon the switch from a $Ca^{2+}$-containing to a DVF $Na^+$-containing solution. UV light-induced currents of Orai1 A137Bpa also exhibited a 2-fold increase in DVF-$Na^+$ versus $Ca^{2+}$ current levels already in the absence of STIM1 (Fig. 8a, b, Supplementary Fig. 10a). In contrast, UV-light-triggered Orai1 L174Bpa currents decreased upon the exchange of a $Ca^{2+}$ by a DVF $Na^+$-containing solution (Fig. 8b, Supplementary Fig. 10b), in line with our findings on various Orai1 GoF mutants[43]. Interestingly, UV light-activated Orai1 A254Azi currents in the absence of STIM1 showed comparable (Fig. 8b) or sometimes slightly enhanced (Supplementary Fig. 10c) current densities in the $Ca^{2+}$-containing versus DVF $Na^+$-containing solution. In the presence of STIM1, all light-sensitive Orai1 mutants showed an increase in the ratio $I_{DVF}/I_{Ca2+}$, both before and after exposure to UV light (Fig. 8b, Supplementary Fig. 10d–f).

FCDI of STIM1/Orai1 currents recorded upon a voltage step to negative potentials showed a decrease in maximum currents within the first 250 ms followed by a reactivation phase over the next 1500 ms using 20 mM EGTA in the pipette. Among UV-mediated currents of light-sensitive Orai1 mutants, Orai1 A137Bpa exhibited most comparable extent of FCDI already in the absence of STIM1 compared to STIM1/Orai1 currents upon application of a hyperpolarizing potential to −70mV (Fig. 8c, d). Notably, two-component fit of the current decay, revealing fast and slow τ values, exhibited slightly lower $\tau_{fast}$ for Orai1 A137Bpa currents compared to STIM1/Orai1 (Supplementary Table 1). In the presence of STIM1, the extent of FCDI and $\tau_{fast}$ were slightly enhanced (Fig. 8d, Supplementary Fig. 10g, Supplementary Table 1). In contrast, FCDI of Orai1 L174Bpa and Orai1 A254Azi currents drastically differed from that of STIM1/Orai1 and Orai1 A137Bpa currents, as they exhibited no FCDI, but reactivation (Fig. 8d, Supplementary Fig. 10h, i). In the presence of STIM1, Orai1 L174Bpa showed FCDI, however with enhanced reactivation compared to STIM1/Orai1 currents, while Orai1 A254Azi retained reactivation (Fig. 8d, Supplementary Fig. 10h, i).

Since, FCDI of Orai1 A137Bpa is most similar to wild-type, we continued to investigate it in the presence of BAPTA (20 mM) in the pipette. The extent of inactivation of STIM1/Orai1 currents is reduced using BAPTA due to local and global $Ca^{2+}$ buffering, versus EGTA, buffering only global $Ca^{2+}$ (Fig. 8c, e). Remarkably, Orai1 A137Bpa currents exhibited enhanced levels of FCDI compared to wild-type using BAPTA (Fig. 8e) and reached comparable levels like Orai1 A137Bpa using EGTA (Fig. 8c–f).

To investigate in more detail this unique behavior when using distinct $Ca^{2+}$ buffering, we recorded FCDI at lower hyperpolarizing potentials (−110mV, −90mV). As expected, clear inactivation is seen for STIM1/Orai1 currents using EGTA, which occurs to a reduced extent using BAPTA. In contrast, UV-activated Orai1 A137Bpa currents showed comparable or even higher inactivation using BAPTA compared to EGTA (Fig. 8c, d, Supplementary Fig. 11a–d, Supplementary Table 1). Using instead of $Ca^{2+}$, DVF solution at the extracellular side significantly reduced the extent of FCDI in all cases (Fig. 8f, Supplementary Fig. 11c, d). This clearly indicates a $Ca^{2+}$-dependent inactivation of Orai1 A137Bpa, but with altered inactivation kinetics. In the presence of STIM1 before and after UV light, FCDI of Orai1 A137Bpa currents was restored to a comparable extent as for wild-type using EGTA, however, with abolished reactivation. Interestingly, using BAPTA left FCDI at comparable levels like for EGTA, in contrast to wild-type currents, indicating unique $Ca^{2+}$-dependent properties of Orai1 A137Bpa (Fig. 8d, Supplementary Fig. 11e, f).

Furthermore, we compared the inhibitory effect of the CRAC channel blocker CM-4620[59] on the three light-sensitive Orai1 mutants, Orai1 A137Bpa, Orai1 L174Bpa and Orai1 A254Azi (Fig. 8g, h). In all cases, we discovered a significant block of the UV-mediated currents upon application of 10 μM CM-4620.

Overall, we discovered that photocrosslinking-induced Orai1 A137Bpa activation triggers $Ca^{2+}$ channel-like activation independent of STIM1, matching CRAC channel hallmarks best.

### Photocrosslinking-induced Orai1 activation is suitable to trigger $Ca^{2+}$-dependent downstream signaling processes

In addition to HEK293 cells, we investigated the functionality of Orai1 A137Bpa and Orai1 A254Azi, which showed highest activity, also in the RBL-2H3 mast cell and Jurkat TIB-152 cell lines. Indeed, patch-clamp experiments revealed that both mutants overexpressed in the two cell lines clearly showed a rapid increase in currents after UV light illumination, with a comparable $V_{rev}$ as observed in HEK293 cells (Fig. 9a–f). This indicates their general applicability to trigger $Ca^{2+}$-dependent downstream signaling using UV light. Interestingly, the current levels were reduced and slow inactivation of Orai1 A254Azi was less pronounced compared to analog experiments in HEK293 cells which likely underlies their reduced expression in these cell types as exemplified for RBL-2H3 cells (Supplementary Fig. 6e).

Consequently, we investigated whether UV light-triggered activation of our photocrosslinking UAA-containing Orai1 mutants is suitable to trigger NFAT translocation to the nucleus in HEK293 cells. UV light-mediated stimulation of all three mutants (Orai1 A137Bpa/L174Bpa/A254Azi) triggered NFAT translocation to the nucleus with significantly higher levels compared to control cells not exposed to UV light or overexpressing Orai1 wild-type (Fig. 9g–i). Remarkably, Orai1 A137Bpa exhibited most pronounced UV-light triggered NFAT translocation. Analogously, we observed UV light-triggered NFAT translocation in RBL-2H3 cells containing Orai1 A137Bpa or Orai1 A254Azi comparable to TG-stimulated Orai1-expressing cells and significantly different to unstimulated cells (Fig. 9 j–l).

Overall, we conclude that photocrosslinking-induced Orai1 activation is generally applicable to various cell lines and is suitable to induce $Ca^{2+}$-dependent downstream signaling.

### Discussion

Using optoproteomics, we generated a family of Orai1 mutants that can be precisely activated by light, independent of STIM1. We showed that

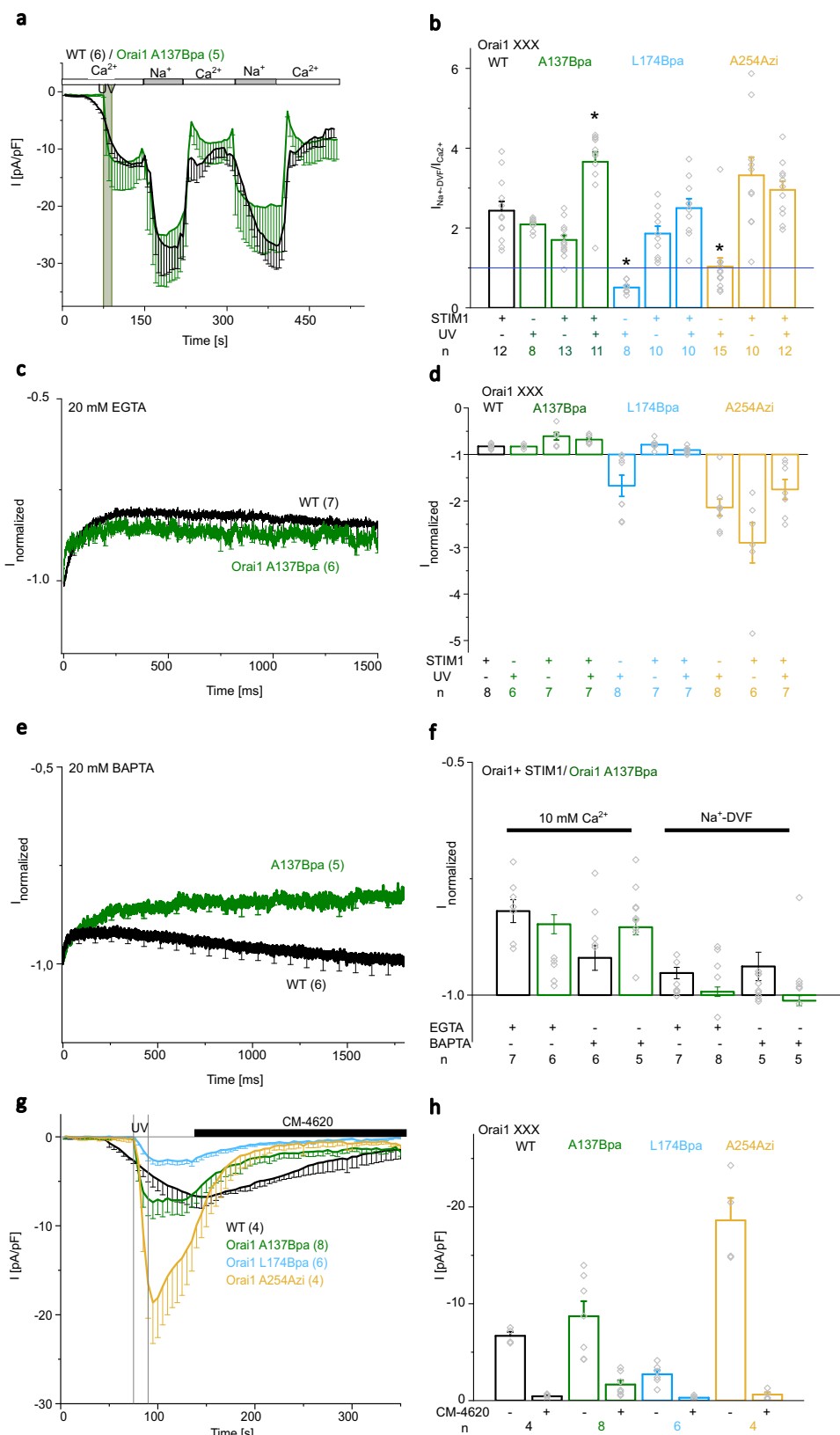

the incorporation of the photocrosslinking UAAs in the different TM domains is highly reliable and enables robust photomodulation with high temporal and spatial precision capable of triggering downstream signaling. Mechanistically, it is conceivable that photocrosslinking-induced covalent bond formation triggers global interdependent TM domain motions which lock Orai1 in a higher or lower active conformation.

Azi and Bpa were incorporated into all tested Orai1 mutants, containing an Amber stop codon (Fig. 1, Fig. S1), as evidenced by preserved plasma membrane localization, underlining the viability and

**Fig. 8 | UV light-activated Orai1 A137Bpa currents match CRAC channel hall-marks best. a** Time course of current densities after whole-cell break-in comparing Orai1 A137Bpa with wild-type STIM1/Orai1, while repeatedly switching from 10 mM Ca$^{2+}$- (I$_{Ca2+}$) to a DVF Na$^+$- (I$_{Na+-DVF}$) containing solution. 15 s UV pulse was only applied to Orai1 A137Bpa. **b** Bar diagram summarizing the ratio of currents (I$_{Na+-DVF}$/I$_{Ca2+}$) for conditions shown in (**a**) and for Orai1 A137Bpa with STIM1 and Orai1 L174Bpa and Orai1 A254Azi in the absence and presence of STIM1 with and without UV light (15 s). Single values are indicated in gray. An increase in the I$_{Na+-DVF}$ was measured when the threshold marked in blue was exceeded (Welch-ANOVA for (**b**): F(9;38,52)=62,85, $p = 0$). **c** Time course showing normalized currents of Orai1 A137Bpa compared to wild-type CRAC channel (STIM1 + Orai1; WT) obtained upon application of a voltage step to −70 mV from a holding potential of 0 mV using 20 mM EGTA in the pipette. **d** Bar diagram summarizing normalized currents at t = 250 ms from data depicted in (**c**) and for other conditions and mutants in analogy to (**b**). Single values are indicated in gray (Welch-ANOVA for (**d**):

F(9;23,63) = 12,79, $p = 3,54*10^{-7}$). **e** Time course showing normalized currents in analogy to (**c**) using 20 mM BAPTA in the pipette. **f** Bar diagram summarizing normalized currents at t = 250 ms from data depicted in (**c**) and (**e**) using either 20 mM EGTA or 20 mM BAPTA in the pipette and 10 mM Ca$^{2+}$ or Na$^+$-containing DVF solution at the extracellular side (Welch-ANOVA for (**f**): F(7;15,82)=13,41, $p = 1,27*10^{-5}$). **g** Time courses of Ca$^{2+}$ current-densities after whole-cell break-in of Orai1 A137Bpa, Orai1 L174Bpa and Orai1 A254Azi exposed to UV-light (15 s) compared to STIM1/Orai1 currents activated by passive store-depletion. After maximal activation the Orai1 channel blocker CM-4620 (10 μM) was applied. **h** Summarizing bar diagram comparing current-densities in (**g**) at maximal levels and after application of CM-4620 (t = 320 s; Welch-ANOVA for (**h**): F(7;12,19)=39,98, $p = 2,2*10^{-7}$). Data represent mean values ± SEM of indicated number (n) of experiments. *Significant differences ($p < 0.05$). Detailed statistic values are shown in Supplementary Table 3. Source data are provided as a Source Data file.

robustness of this technology. Though Azi and Bpa moieties are bulkier than canonical amino acids, our high incorporation efficiency argues for their overall good tolerability. Despite the fact that Bpa exhibits more specific photochemistry and enhanced stability compared to Azi[60], incorporation of both at several positions altered activity upon UV light irradiation. Maximum activation of photoresponsive Orai1 mutants, already active independent of STIM1, reached levels comparable to or even higher than those of STIM1-activated Orai1. Despite the fact that Bpa is bulkier and has less mobility to rotate and crosslink with proximal hydrocarbons than Azi[61], it allowed UV-induced activation at similar amount of sites like Azi. Positions enabling photomodulation are fairly evenly distributed in all TM domains, suggesting, in agreement with previous studies[13,14,39,52], that Orai1 pore opening is fine-tuned by a series of checkpoints. The sites which allowed strong UV light-induced activation are located close to and within the cytosolic helical extension of the TM domains (Fig. 1d). This suggests that the incorporation of bulky photocrosslinking UAAs in this area together with UV light excitation stabilizes Orai1 in an open state, structural effects that are likely also induced by STIM1 coupling. Collectively, the use of light-sensitive UAAs opens up a new dimension allowing remote real-time and dynamic monitoring of CRAC channel structure and function at the amino acid level, which overcomes the limited resolution of traditional technologies. Together with conventional mutagenesis this method will allow to resolve inter-TM domain motions required for Orai1 pore opening.

The incorporation of Azi or Bpa allowed to monitor changes in Orai1 activity that were tightly coupled in time to the illumination events. Remarkably, the overall function and especially the gating machinery of the Orai1 Ca$^{2+}$ channel were hardly affected by the incorporation and excitation of the photocrosslinking group. Azi and Bpa thus enable an artificial control mechanism that depends on different amino acids and is orthogonal to the evolutionarily intended natural mechanism (STIM1 binding) - a promising feature for biological studies. We have shown that UV-triggered activation of our three light-sensitive Orai1 mutants occurred independently of STIM1. Low UV light-induced currents in KO cells, RBL-2H3 and Jurkat T cells, and less pronounced slow Ca$^{2+}$-dependent inactivation of Orai1 A254Azi in RBL-2H3 and Jurkat T cells, are most likely attributable to lower expression levels. Noteworthy mammalian cells studied here also express other Orai isoforms, which could lead to the formation of heteromeric complexes and impact UV-mediated current properties. Indeed, cells with different ratios of co-expressed Orai1 and Orai1 A137Bpa exhibited higher currents the less Orai1 was present. However, we observed comparable V$_{rev}$ in different mammalian cells and overexpressed proteins are typically present at much higher levels compared to endogenous proteins. Nevertheless, the effects of heteromer formation in RBL-2H3 and Jurkat T cells cannot be excluded.

Interestingly, UV-light activated Orai1 A254Azi currents exhibited Ca$^{2+}$ dependent slow inactivation after maximal activation. However,

the molecular determinants of this remain to be elucidated. To date, the slow Ca$^{2+}$-dependent inactivation of CRAC currents has been associated with the accessory protein SARAF, which facilitates the return of STIM1 to the resting state[62,63] via a complex SARAF-STIM1 interaction mechanism.

Despite photocrosslinking-induced activation of the respective Orai1 UAA-containing mutants being independent of STIM1, physiological STIM1-mediated activation of Orai1 A137Bpa, Orai1 L174Bpa and Orai1 A254Azi occurred to similar levels like for wild-type Orai1. This underlines that UAA incorporation per se does not affect the evolutionary defined pore opening mechanism. Interestingly, STIM1 binding affects UV-mediated activation of Orai1 A137Bpa, but not Orai1 L174Bpa or Orai1 A254Azi, in an inhibitory manner in contrast to only UV-activated UAA-containing Orai1 mutant. This indicates that the area around A137Bpa, in particular, the cytosolic extension of TM2 is critical for communication with STIM1, either in a direct or allosteric manner.

In contrast to STIM1, a number of GoF mutations in all TM domains impaired photocrosslinking-induced pore opening. These inhibitory effects are likely due to conformational changes nonpermissive for pore opening and/or interfering with photocrosslinking. Alternatively, since some constitutive Orai1 double mutants showed only UV-mediated current enhancements in the absence, but not in the presence of STIM1, we hypothesized that the combination of the GoF mutation with either UV light or STIM1 is sufficient in these cases for maximum activation. Only Orai1 ANSGA or weakly active Orai1 GoF mutants (V181A[14,52], P245L[40] (in some cases)) allowed additional photocrosslinking-induced activation independent of the presence of STIM1, suggesting that they do not or marginally interfere with the photo-crosslinking-triggered and even physiological signal propagation. Other GoF mutants likely adopt distinct conformational pathways leading to pore opening.

In the absence of any other activating factor such as STIM1 or a GoF mutation, all three photocrosslinking activatable Orai1 variants exhibit comparable key gating sites like STIM1-activated Orai1 including the selectivity filter[54], an intact N-terminus[43,51,55,64] and the cytosolic triangles formed by salt-bridge and hydrophobic interactions[14,53] within the cytosolic portions. Investigation of the three light-sensitive Orai1 variants revealed, that Orai1 A137Bpa matched with typical CRAC channel characteristics best. While all mutants showed a V$_{rev}$ in the range of +50 mV and were inhibited by a CRAC channel blocker, only Orai1 A137Bpa exhibited enhancement in I$_{DVF}$ versus I$_{Ca2+}$ and FCDI. Interestingly, the extent of FCDI of UV light-activated Orai1 A137Bpa in response to distinct Ca$^{2+}$ buffering differed compared with STIM1/Orai1 wild-type currents. Nevertheless, FCDI of Orai1 A137Bpa reduced significantly in the presence of DVF Na$^+$-containing solution at the extracellular side, increased with enhancing hyperpolarizing voltage, and showed a biphasic behavior as the wild-type. This indicates that Orai1 A137Bpa shares the basic mechanism of Ca$^{2+}$-dependent inactivation gating with STIM1/Orai1. Similar behavior was previously

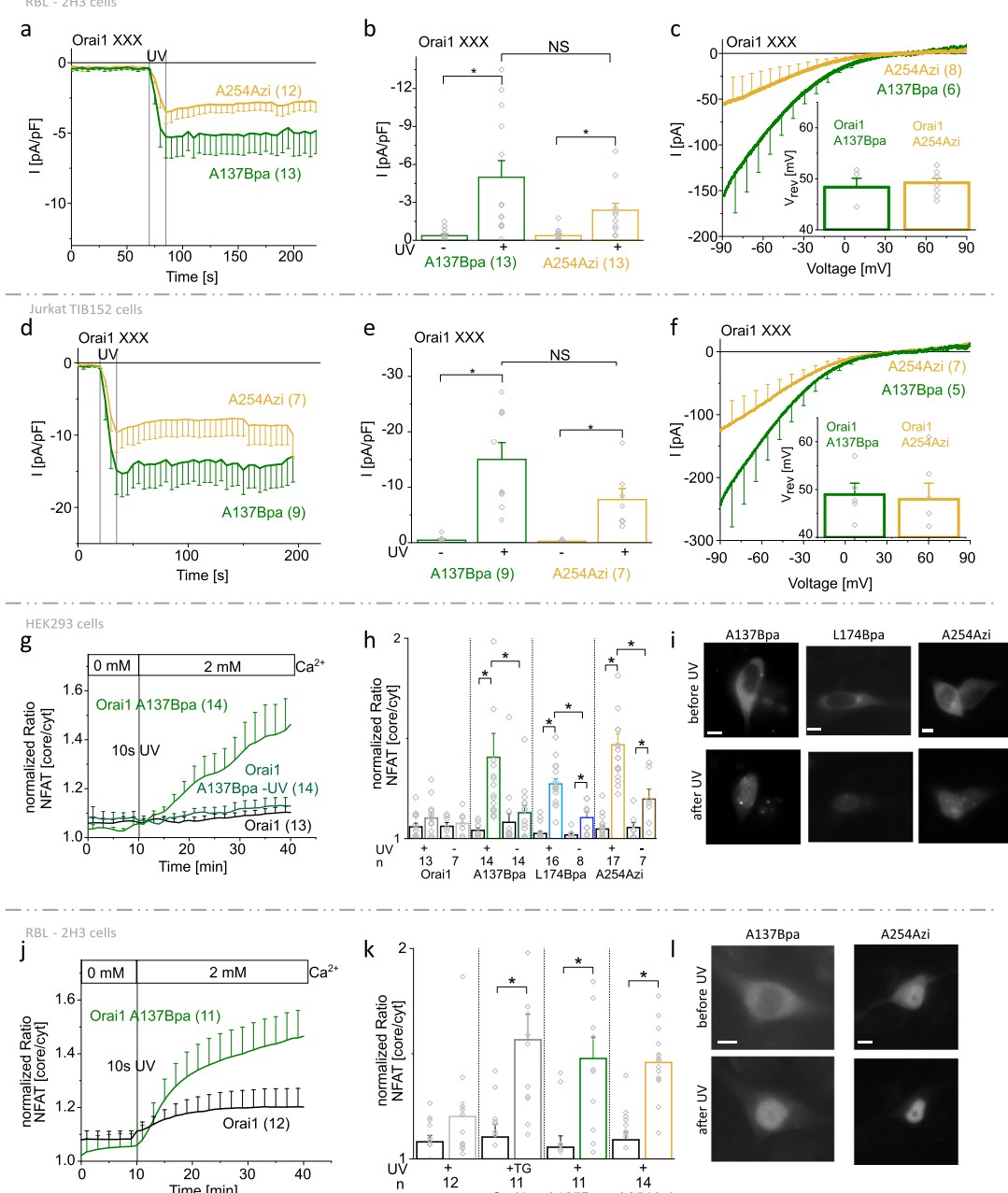

**Fig. 9 | Photocrosslinking of light-sensitive Orai1 mutants trigger $Ca^{2+}$ -dependent downstream signaling.** Time course of current densities after whole-cell break-in comparing Orai1 A137Bpa and Orai1 A254Azi in RBL-2H3 cells (**a**) and Jurkat TIB-152 cells (**d**). **b**, **e** Corresponding bar diagram to (**a**) and (**d**), respectively. Values taken from time points 25 s and 125 s (**a**) and 15 s and 75 s (**d**), respectively. **c**, **f** Corresponding I/V relationships were taken at 125 s in (**a**) and 75 s in (**d**). Inlet represents reversal potential ($V_{rev}$) of light-sensitive Orai1 mutant currents. Time course of NFAT translocation into the nucleus (normalized ratio NFAT (core/cytosol (cyt)) indicating the fluorescence ratio nucleus:cytosol) of Orai1 A137Bpa with and without (only in (**g**)) application of UV light compared to wild-type Orai1 in HEK293 (**g**) or RBL-2H3 (**j**) cells. NFAT translocation was monitored initially in 0 mM $Ca^{2+}$ solution, followed by 2 mM $Ca^{2+}$ solution after 10 min together with the application of UV light (10 s). **h**, **k** Bar diagram summarizing the extent of NFAT translocation for wild-type Orai1, Orai1 A137Bpa, and additionally Orai1 L174Bpa and Orai1 A254Azi with (**h**) & (**k**) and without application of UV light (only in (**h**)) and thapsigargin-activated (+TG) Orai1 expressing cells (only in (**k**)). Paired bars show the ratio under resting (black, t = 3 min) and activated (colored, t = 35 min) conditions. **i** and **l** Fluorescence images of CFP-labeled NFAT of representative cells corresponding to (**g**), (**h**), (**j**) and (**k**) before and after application of 10 s UV light. White bars indicate 5 μm. Data represent mean values ± SEM of indicated number (*n*) of experiments. *Significant differences ($p < 0.05$) tested by Mann–Whitney test. Detailed statistic values are shown in Supplementary Table 3. Source data are provided as a Source Data file.

reported for a mutation of T92 (Orai1 T92W), a position located in close proximity of 6 Å to A137[65]. Enhanced FCDI using global and local $Ca^{2+}$ buffering has been supposed to underlie enhanced $Ca^{2+}$ sensitivity of Orai1 T92W compared to Orai1 wild-type[65]. Previously, the inhibitory domain (aa470-491) of STIM1 was shown to be essential for FCDI[66,67].

Indeed, STIM1 restored the extent of FCDI of Orai1 A137Bpa to comparable levels as for wild-type, though reactivation was abolished and distinct buffering conditions did not alter the extent of FCDI. Moreover, Orai proteins contain regions that modulate FCDI, including the N-terminus[66,68–70], loop2[71] and the C-terminus[72,73]. Notably, the basic

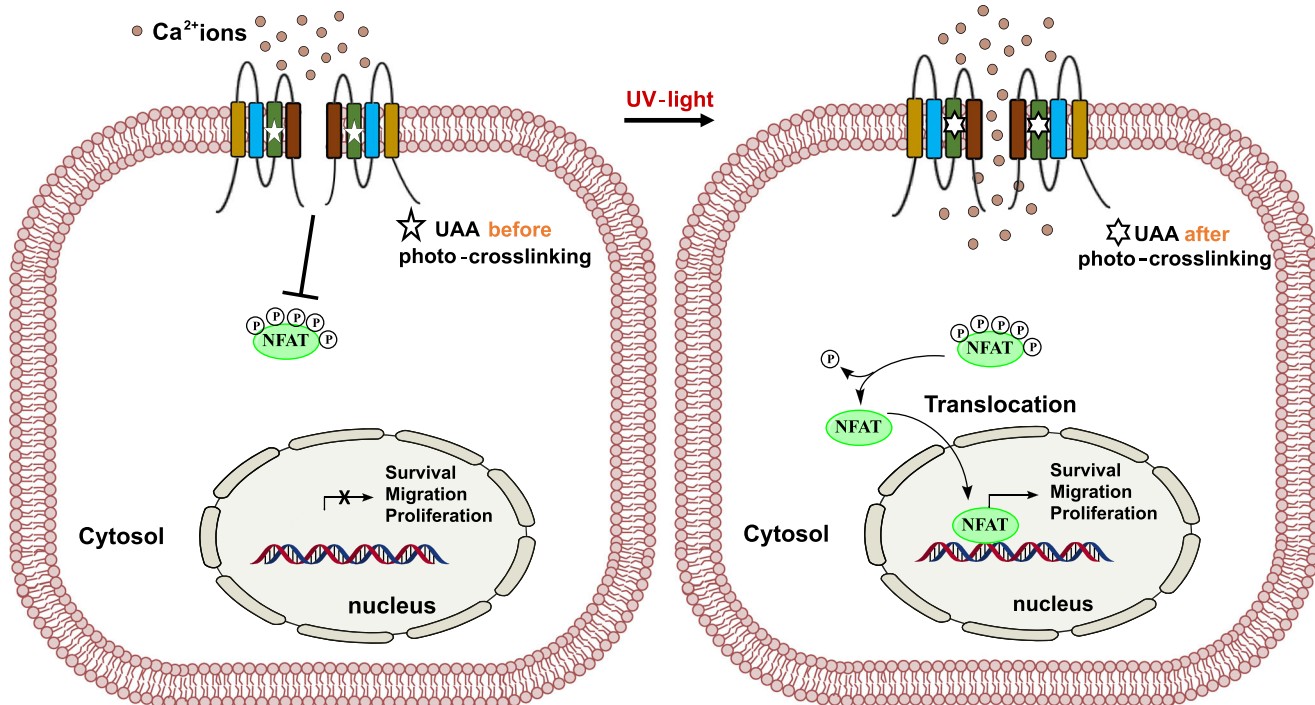

**Fig. 10 | Photocrosslinking-induced Orai1 activation leads to CRAC channel-like Ca²⁺ influx suitable to trigger nuclear factor of activated T-cells (NFAT) translocation.** (left) Closed Orai1 channel (represented by two subunits containing four transmembrane (TM) domains) which has a photocrosslinking UAA (star) in one of the TM domains incorporated. (right) Upon application of UV light, photocrosslinking triggers a local conformational change, which is transferred as a global structural change to the entire channel complex and triggers pore opening. Subsequent Ca²⁺ influx leads to the activation of downstream signaling cascades such as the nuclear translocation of NFAT.

pore segment (aa76-91) is involved in FCDI[70], which is in direct contact with T92W and may interplay with A137Bpa. This strengthens the hypothesis that the inner pore of Orai1 is an essential factor in mediating FCDI. Overall, these results suggest that FCDI is an intrinsic property of Orai channels that is further tuned by STIM1 coupling to Orai1 C-terminus, but possibly also to other cytosolic segments. Our findings open avenues for characterizing the key determinants mediating inactivation and Ca²⁺-sensing of Orai1.

In essence, we employed a powerful method to transfer light-sensitivity to Orai1 to remotely and tightly control Ca²⁺-dependent downstream signaling (Fig. 10). We demonstrated that instead of STIM1, photocrosslinking of TM domains at selected positions is sufficient to induce pore opening. The library of light-sensitive Orai1 mutants will be valuable to dissect structural features determining biophysical properties along with still unknown binding interfaces of CRAC channels, which will stimulate structural and computational studies. In vivo, this currently emerging technology holds the potential to provide fundamental insights into the biology of native ion channels and uncover new targets beneficial for human therapy.

## Methods
### Molecular biology
For C-terminal fluorescence labeling of human Orai1 (Orai1; accession number NM_032790, provided by the laboratory of A. Rao), the construct was cloned into the pEYFP-N1 (Clontech) expression vector via XhoI and BamHI restriction sites. Orai1 N-terminal deletion mutants (Orai1 Δ1–47, Δ1–60, Δ1–64, Δ1–68, Δ1–70, Δ1–71, Δ1–72, and Δ1–78) were amplified via PCR, including an N-terminal EcoRI and a C-terminal SacII restriction site. Site-directed mutagenesis (V102A, H134A, V181K, V181A, P245L, ₂₆₁ANSGA₂₆₅ (L261A-V262N-H264G-K265A), E106Q, K85E, E149K, L174D, A235C, L185A, L273D) and the introduction of the TAG stop codon within the Orai construct for the incorporation of the UAA at the sites of interest was performed using the QuikChangeTM

XL site-directed mutagenesis kit (Stratagene) with the corresponding Orai1 construct serving as a template. Primers are listed in Supplementary Table 2.

Human STIM1 (STIM1; accession number NM_003156) N-terminally ECFP-tagged was kindly provided by the laboratory of T. Meyer (Stanford University). Site-directed mutagenesis (L373S and L373S A376S) was performed using the QuikChangeTM XL site-directed mutagenesis kit (Stratagene) with the corresponding STIM1 construct serving as a template. The integrity of all resulting clones was confirmed by sequence analysis (Eurofins Genomics/Microsynth).

The Calcium indicator R-GECO1.2 was purchased from Addgene (#45494[74]) as well as ER-expressing LAR-GECO1 (Addgene: #61244[75]) and the humanized versions of the aminoacyl-synthetase/tRNA pairs recognizing azido-L-phenylalanine and benzoyl-L-phenylalanine (Addgene: #105829[36] and #155342[35]). CFP-NFAT was kindly provided by R. Kehlenbach (Scripps Research Institute).

### Cell culture and transfection
Human embryonic kidney 293 (HEK293) (#ACC305) purchased from DSMZ (German Collection of Microorganisms and Cell Culture GmbH), CRISPR/Cas9 STIM1/Orai1 DKO HEK293[76], CRISPR/Cas9 STIM1/STIM2 DKO HEK293[77], CRISPR/Cas9 Orai1/Orai2/Orai3 TKO HEK293[78], rat basophilic leukemia (RBL-2H3) cells and Jurkat (ATCC TIB-152) cells were cultured in DMEM, MEM and RPMI-1640, respectively, as recommended by the DSMZ or as stated in the respective references. All media were supplemented with 1-glutamine (2 mM), streptomycin (100 µg/ml), penicillin (100 units/ml) and 10% fetal calf serum while growing at 37 °C in a humidity-controlled incubator with 5% CO₂. CRISPR/Cas9 STIM1/Orai1 DKO HEK293 cells were kindly provided by Rajesh Bhardwaj and Matthias A. Hediger[76]. CRISPR/Cas9 STIM1/STIM2 DKO HEK293 cells were kindly provided by Mohammed Trebak[77], CRISPR/Cas9 Orai1/Orai2/Orai3 TKO HEK293 cells were kindly provided by Barbara Niemeyer[78].

For all HEK293 cell lines, transient transfection was performed[79] using TransFectin™ Lipid Reagent (Bio-Rad Laboratories, Inc.; 2 µl per transfection). The plasmid ratio used in the experiments was 1 µg Orai1: 2 µg tRNA/aaRS pair for normal HEK293 cells. In the presence of STIM1 1 µg STIM1, for Ca²⁺ imaging 1 µg R-GECO1.2 and for NFAT experiments 1 µg NFAT were used. To circumvent lower expression levels (indicated in Supplementary Figure 6e), in both DKO HEK293 cell lines the amount of Orai1 plasmids was raised to 1.5 µg. Experiments were performed 24 h after transfection in normal HEK293 or after 48 h for experiments in CRISPR/Cas9 Orai1/Orai2/Orai3 TKO HEK293. RBL-2H3 cells and Jurkat-TIB152 cells were transfected via electroporation using the GenePulser Xcell (Bio-Rad Laboratries, Inc.) (exponential protocol; 950µF; 250 V; 0.4 cm cuvette and 400 µl cell suspension in their respective media). 10 µg Orai1: 10 µg STIM1: 20 µg tRNA/aaRS: 5 µg NFAT was used. Throughout the manuscript, we used Orai1 wild-type/mutant-YFP and where applicable STIM1-CFP or NFAT-CFP. Growth media of all transfected cells was supplemented with the specific UAA (1 mM; azido-L-phenylalanine and benzo-L-phenylalanine, BACHEM, dissolved in 0.5 M NaOH). Potential mycoplasma contamination was checked regularly using VenorGeM Advanced Mycoplasma Detection Kit (VenorGeM). CM-4620 was purchased at MedChemExpress.

## Calcium imaging

HEK293 cells, transfected with above mentioned ratios of plasmids, were grown on coverslips for 1 day. Coverslips were transferred to an extracellular solution without Ca²⁺ and mounted on an Axiovert 135 inverted microscope (Zeiss, Germany) equipped with a sCMOS-Panda digitale Scientific Grade camera 4.2 MPixel and a LedHUB LED Light-Engine light source (LedHUB®; Omicron-Laserage Laserprodukte GmbH). Excitation of R-GECO1.2 was obtained using the LED spanning 500 and 600 nm together with a Chroma filter allowing excitation between 540 and 580 nm and emission between 590-660 nm. UV light excitation was obtained using the 365 nm LED. Ca²⁺ measurements are shown as normalized intensities of R-GECO1.2 fluorescence in HEK293 cells. Due to filter exchange, R-GECO1.2 or LAR-GECO1 intensity could not be recorded simultaneously with exposure to UV light. During the Ca²⁺ imaging experiments the fluorescence intensity of R-GECO1.2 or CFP-NFAT was recorded every 10 s or 10 min, respectively. Image acquisition and intensity recordings were performed with Visiview5.0.0.0 software (Visitron Systems). A Thomas Wisa perfusion pump was used for extracellular solution exchange during the experiment. All experiments were performed at room temperature using extracellular solutions containing (in mM): 140 NaCl, 10 HEPES, 10 glucose, 5 KCl, 1 MgCl₂, pH 7.4 and 0/0.5/2 CaCl₂, respectively. In the initial screen of light-sensitive Orai1 mutants (Fig. 1d and Supplementary Fig. 1) 0.7 mW/cm² UV light and 0.1 mW/cm² for excitation of R-GECO1.2 were applied. Control experiments with varying excitation strength for detection of R-GECO1.2 (Supplementary Fig. 3a–c) showed maximal Ca²⁺ levels between 0.3 and 0.7 mW/cm², while increasing UV intensities (Supplementary Figure 3d) showed comparable maximum Ca²⁺ levels. Hence, we applied throughout the manuscript (Figs. 2–8; Fig. S4–11) 0.7 mW/cm² UV light and 0.3 mW/cm² for excitation of R-GECO1.2.

## Electrophysiology

The electrophysiological setup consisted of an inverted microscope (Zeiss Axiovert 200) combined with the Axopatch 200B amplifier (Molecular Devices), the Scientifica PatchStar micromanipulator and the light engine (Lumencor Spectra III). pClamp11 was used for electrophysiological recordings. Control experiments varying the applied UV intensity (Supplementary Fig. 3i–k) resulted in fast maximum activation starting at 2.2 mW/cm² UV intensity, thus, this intensity was used within all experiments. Electrophysiological experiments were performed at room temperature using whole cell configuration and an Ag/AgCl reference electrode. Cells grown in petri dishes were reseeded

18 h or 42 h in 0.1 mM Ca²⁺ containing medium, respectively after transfection on poly-L-lysine treated coverslips. Patch-Clamp experiments were performed 6–10 h after reseeding. For time-course and I/V measurements, voltage ramps were applied every 5 s ranging from −90mV to +90 mV over 1 s starting from a holding potential of 0 mV. To determine FCDI, voltage steps were applied to −70mV/−90mV/−110mV for 2000ms starting from a holding potential of 0 mV. Passive store-depletion was initiated by the internal pipette solution (in mM): 145 Cs methane sulfonate, 20 EGTA or 20 BAPTA, respectively, 10 HEPES, 8 NaCl, 3.5 MgCl₂, pH 7.2. Standard extracellular solution contained (in mM): 145 NaCl, 10 HEPES, 10 CaCl₂ (or 0.5; 1; 1.5; 2; 10; 20; 110 in Supplementary Fig. 4e), 10 glucose, 5 CsCl, 1 MgCl₂, pH 7.4. Na⁺-DVF solution contained (in mM): 150 NaCl, 10 HEPES, 10 glucose, 10 EDTA, pH 7.4. A Thomas Wisa perfusion pump was used for extracellular solution exchange during the experiment. Applied voltages were not corrected for the liquid junction potential, which was determined as +12 mV. All currents were leak subtracted either by subtraction of the initial current trace immediately after whole cell break-in with no visible current activation, or of a 10 µM La³⁺-blocked current traces at the end of the experiment. The time course of FCDI was fit with a double-exponential function and the fast and slow time constants ($\tau_{fast}$, $\tau_{slow}$) were determined from the fits. The reactivation and slow inactivation were fit with a single exponential function and τ was determined from the fits. Complex inactivation behaviors combining FCDI and reactivation were investigated by piecewise fit combining a single- or double exponential function with a linear function, of which τ was determined for the FCDI part.

## Confocal FRET fluorescence microscopy

Confocal FRET microscopy was carried out at room temperature 18–24 h after transfection. The standard extracellular solution contained (in mM): 145 NaCl, 5 KCl, 10 HEPES, 10 glucose, 1 MgCl₂, 2 CaCl₂ and was set to pH 7.4. For Ca²⁺ store depletion, a Ca²⁺-free extracellular solution containing 1 µM thapsigargin or 10 µM BHQ was used. The experimental setup consisted of a CSU-X1 Real-Time Confocal System (Yokogawa Electric Corporation, Japan) combined with two CoolSNAP HQ2 CCD cameras (Photometrics, AZ, USA). The installation was also fitted with a dual port adapter (dichroic, 505lp; cyan emission filter, 470/24; yellow emission filter, 535/30; Chroma Technology Corporation, VT, USA). An Axio Observer.Z1 inverted microscope (Carl Zeiss, Oberkochen, Germany) and two diode lasers (445 and 515 nm, Visitron Systems, Puchheim, Germany) were connected to the described configuration. All described components were positioned on a Vision IsoStation antivibration table (Newport Corporation, CA, USA). A perfusion pump (ASF Thomas Wisa, Wuppertal, Germany) was used for extracellular solution exchange during experiments. Image recording and control of the confocal system were carried out with the VisiView software package (v.2.1.4, Visitron Systems). The illumination times for individual sets of images (CFP, YFP, FRET) that were recorded consecutively with a minimum delay were kept in a range of 100–300 ms. Due to cross-excitation and spectral bleed-through, image correction before any FRET calculation was required. YFP cross-excitation (*a*) and CFP crosstalk (*b*) calibration factors were therefore determined on each measurement day using separate samples in which cells only expressed CFP or YFP proteins. FRET analysis was limited to pixels with a CFP:YFP ratio between 0.1:10 and 10:0.1. After this threshold determination as well as background signal subtraction, the apparent FRET efficiency $E_{app}$ was calculated on a pixel-to-pixel basis. This was performed with a custom program integrated into MATLAB (v.7.11.0, The MathWorks, Inc., MA, USA) according to the following equation

$$E_{app} = \frac{I_{FRET} - aI_{YFP} - bI_{CFP}}{I_{FRET} - aI_{YFP} + (G - b)I_{CFP}}$$

where $I_{FRET}$, $I_{YFP}$ and $I_{CFP}$ denote the intensities of the FRET, YFP and CFP images, respectively. $G$ denotes a microscope-specific constant parameter that was experimentally determined as 2.75[80].

## Statistics and reproducibility

Results are presented as mean value ± SEM calculated for the indicated number $n$ of experiments. For statistical comparison, the Mann-Whitney test was performed for comparison of two independent samples considering differences as statistically significant at $p < 0.05$. Levene test was used to test for variance homogeneity. If fulfilled, one-way ANOVA test was used for statistical comparison of multiple independent samples using the F-distribution. If not fulfilled, the Welch-ANOVA test was used instead. Subsequently, Fisher's least significant post-hoc test was used after one-way ANOVA, while Games-Howell post hoc test was used after Welch-ANOVA to determine the pairs that differ statistically significant ($p < 0.05$). In all cases, the Shapiro-Wilk test was applied to prove normal distribution of the respective datasets (or one-sample Kolmogorov-Smirnov test). F and p values of statistics performed are included in Supplementary Table 3 for all samples.

All $Ca^{2+}$ imaging experiments were performed on three days in paired-comparison leading to similar results. All electrophysiological experiments were carried out at least on two different days in paired comparison leading to similar results. Cell images were taken for each experiment showing comparable cellular distribution of the respective proteins/mutants, as shown in one representative image in the respective figures.

## Reporting summary

Further information on research design is available in the Nature Portfolio Reporting Summary linked to this article.

## Data availability

Graphical illustration of a modeled human Orai1 structure are based on 4HKR (https://doi.org/10.2210/pdb4HKR/pdb). The source data underlying Figs. 1–10, and Supplementary Fig. 1–11 are provided as a Source Data file and are deposited in an open public repository [https://doi.org/10.5281/zenodo.7551827] Source data are provided with this paper.

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

## Acknowledgements

We thank S. Buchegger for excellent technical assistance. This work was
supported by the Austrian Science Fund (FWF) projects P30567, P32851,
P35900 and P36202 to I.D., P32947 to M.F., P32075-B to I.F.. L.M. holds a
PhD scholarship of Upper Austria within the FWF W1250-B20 Upper Austria
DK NanoCell Project. H.G. holds a PhD scholarship of the Austrian Science
Fund (FWF) PhD program W1250 NanoCell. D.K. is a member of the PhD
program (DK) "Metabolic and Cardiovascular Disease" (FWF W1226).

## Author contributions

L.M., S.W., H.N. and I.D. conceived and coordinated the study and wrote
the paper. L.M., S.W., H.N., M.L., C.H. performed and analyzed electro-
physiological experiments. S.W. and L.H. performed and analyzed $Ca^{2+}$
imaging experiments. M.S., H.G. and D.K. carried out fluorescence
microscopy experiments. H.N., S.L., S.B., V.H., A.B., M.F., I.F., A.T. con-
tributed to molecular biology and biochemistry. All authors reviewed
the results and approved the final version of the manuscript.

## Competing interests

The authors declare no competing interests.

## Additional information

**Supplementary information** The online version contains
supplementary material available at

Isabella Derler.

**Peer review information** *Nature Communications* thanks the anon-
ymous reviewers for their contribution to the peer review of this
work. Peer reviewer reports are available.

