## [Peer Review File · Nature Communications]

Photocrosslinking-induced CRAC channel-like Orai1 activation independent of STIM1Reviewers' Comments:

Reviewer #1:

Remarks to the Author:

In this study Maltan et al reports successful incorporation of the photoreactive unnatural amino acids (UAA), p-benzoyl-L-phenylalanine (Bpa) and p-azido-L-phenylalanine (Azi), at three strategic positions A137, L174 and A254 located in the second, third and fourth transmembrane regions of the Orai1 channel, respectively. They show that illumination of UV light onto cells expressing either one of the engineered channels induces a strong calcium flux independently of ER calcium depletion or co-expression of STIM1. Out of the three UV activated currents, the one induced by A137Bpa exhibits several of the key characteristics of the STIM1 activated Orai1 current suggesting that a similar active conformation of Orai1 is stabilized under both conditions. Further in line with a significant similarity between the STIM1 and UV induced Orai1 currents, UV-induced Ca²⁺ flux in Jurkat or RBL cells activates downstream signaling through the calcineurin-NFAT pathway. Hence, the authors generated a novel method for UV induced activation of Orai1 channels in a manner that appear to closely mimic the native activation mechanism by STIM1.

Overall, the study represent a major technological breakthrough, the experiments are well planned and executed and the main conclusions of the work are (for the most part) supported by wealth of new data. My assessment is that this is an important study that will likely be of interest to a broad scientific community. There are only a few points that should be addressed prior to publication of the manuscript.

1. The analysis of FCDI by the different engineered Orai1 channels is lacking. This type of CDI is usually characterized by two (fast and slower) decay constants and is sensitive to mutations in either STIM1 or Orai1. The traces shown in figure 7A suggest that while the overall magnitude of inactivation by the native (STIM1-Orai1) and artificial (Orai1A137Bpa) is similar the kinetics of inactivation may be different. Please include analysis of the respective tau's for each of the Orai1 mutants. Also, it is not clear that the decay in Orai1 A137Bpa (or in the other mutants) current represent CDI as the control experiment using high levels of BAPTA for fast chelating of incoming Ca²⁺ is missing. Please provide this control for Orai1 A137Bpa. Finally, there is no explanation for how FCDI is maintained in Orai1 A137Bpa without STIM1. What enables the Ca²⁺ sensitivity of the process? Please discuss these issues.

2. There are interesting cell type specific effects regarding current magnitude and inactivation of Orai1 A137Bpa and A254Azi. For example, UV induced currents mediated by Orai1 A254Azi show a prominent slow current inactivation in HEK293 cells, a more moderate one in Jurkat cells while the inactivation is lacking in RBL cells. Also, the relative UV-induced current magnitude of Orai1A137Bpa and Orai1A254Azi is different in the different type of cells. There is no reference or discussion of these effects. Is the inactivation of Orai1 A254Azi calcium dependent? Why is inactivation lacking only in RBL cells? What may lead to these differences in current densities or inactivation of the two engineered channels in the different type of cells? These issues deserve more attention since in addition to STIM1 there are several other proteins like CRACR2A, SARAF or STIMATE that could affect Orai1 function.

3. In Figure 5a – results indicate no change in current magnitude upon UV illumination of the double mutants harboring the V102A mutation. Since STIM1 binding to this Orai1 mutant alters ion selectivity it is important to know whether UV illumination exerts the same effect.

4. The manuscript will benefit from additional editing to correct for awkward phrasings, grammar issues and typos (including reference duplication). More specifically, please work harder to tighten the discussion section, omit sections in which results are simply restated and replace them with new ones that address how and to what purpose the UV-induced Orai1 activation presented in this study may be used in future studies.

Reviewer #2:

Remarks to the Author:

Maltan et al, describe how a set of UV-photocrosslinking unnatural amino acids (UAA) incorporated at different positions of the TM domains of ORAI1 channel can gate the channel in a manner resembling physiological stimulation by STIM1. Authors screen different mutants with two types of modifications, p-benzoyl-L-phenylalanine (Bpa) and p-azido-L-phenylalanine (Azi), ending up with 3 mutants, each in a different TM (TM2, 3 and 4) that allow Orai1 opening upon UV exposition. Two of the mutants (A137Bpa and S254Azi) also display small basal Ca²⁺ entry, similar to Orai1 WT. Authors validate channel activity with Ca²⁺ imaging using the R-Geco2.1 probe, and whole-cell patch-clamp recordings, the two major techniques used in this work. Experiments are well done, and data are convincing, however despite all the hard work and efforts that one can appreciate, I do not see a relevant biological question, neither asked nor answered in the paper. For example, the authors characterize and describe their "tool" comparing it to STIM1 endogenous gating, Orai1 GoF and LoF mutations and N-terminal truncations to end up exploring NFAT translocation as a proxy of signaling. To my understanding they fail to go beyond, after validation of the tool, to describe for example a novel sequential step on the intrinsic Orai1 activation rearrangement, or to convince that the UV-photocrosslinking Orai1 could be a valuable tool to study signaling pathways in physiological context. More specifically:

Figures 1 and S1

1. On Fig 1, the small check marks refer to "no significant change in activity of Orai1 mutants compared to Orai1 wild-type before UV light exposure, but above threshold line". From Fig S1, both A254Azi and A137PBpa mutants, but specially A137Bpa have a strong basal Ca²⁺ entry before UV illumination. That actually is in line with the traces shown of Fig 2. Thus, I don't understand the criteria to classify large or small basal activity of UAA mutations before UV, for me A137Bpa has a strong basal Ca²⁺ entry (more than 1.3 intensity of R-Geco1.2, not at all comparable to Orai1 WT). The criteria must be understandable. Furthermore, what is the threshold line of Fig S1? The asterisks for the significance are not clear. Significant compared to what? What the different asterisk colors refer to? P values should be provided.
2. Figure 2: Orai1 A137Bpa and S254Azi expressing cells display a basal Ca²⁺ entry after switching to 2 mM Ca²⁺ (panels a & c). However, a basal current is not visible before UV illumination on whole-cell recordings for the same UAA modified channels (panels d & f), while the recordings are in 10 mM Ca²⁺. How to interpret this lack of basal current? Did the "leak" disappear because of current subtraction? Is the basal Ca²⁺ entry also observed in absence of STIM1 (STIM1-Orai1 DKO cells)?
3. Why the 10% UV light recordings on Fig. S3 panels abc have such a reduced 2mM calcium response before UV? This should be comparable to all conditions, right? It is difficult to conclude whether the cells had a decreased intrinsic activity that could hardly be further enhanced with UV, or whether the small response to 10% UV really reflect the response to a reduced stimulation?
4. p.5 last sentence: "This indicates efficient UAA incorporation within Orai1 A137Bpa-hexamers" derived from S5, is totally speculative, authors cannot conclude the stoichiometry of the channel based on their analysis. Could the authors discuss how they reached such a conclusion? Otherwise, I would suggest being more descriptive and less conclusive.
5. Fig 4 e,f is hard to understand, and the relevance of such experiments is questionable. On panel f, the light green bars should be labelled STIM1 + and UV -, instead of STIM1 - and UV-; correct? In addition, the condition Orai1 A137Bpa without STIM1 after store depletion and 2mM Ca²⁺ readdition seems to induce larger Ca²⁺ entry than without store depletion (Fig 2a). Is that an effect of STIM2?
6. Fig 5: what is the Erev of Orai1 UAA V102A? Are they non-selective channels like the ones with V102A alone? According to current amplitude versus Ca²⁺ elevation (Fig. 5a&b) it seems to be the case, in particular for the L174Bpa mutant, as the current is very large compared to Ca²⁺ amplitude. Does the expression of STIM1 modifies Erev when the UAA mutations are present together with V102A?
7. p.9 1st paragraph: "all other GoF mutations in the different TM domains, except the slight constitutively active V181A mutant, drastically interfered either with the general function or at least with photocrosslinking-induced activation". I disagree, as the P245L mutation did not interfere with UV activation of all 3 UAA mutants, as shown on Fig 5a,b.
8. Fig 7: FCDI data are not convincing. Why was the voltage step to -70 mV instead of -100 mV or even -120 mV that better reveal the current inactivation? Panel b shows normalized current after

reaching a plateau. But as the current reactivates after the initial 250 ms, what is the plateau referred to? At which time point? It is rather unexpected that FCDI is present in absence of STIM1 for the A137Bpa mutant channel, as an essential part of FCDI is provided by a STIM1 sequence 475-483. This must be discussed.

9. Fig 8: on RBL-2H3 cells, the current of A254Azi is much smaller compared to A137Bpa current, which is very different to what was obtained on HEK cells (A254Azi current is twice as large as A137Bpa current). Is that linked to channel expression? On the other side, the Ca²⁺ selectivity of the two constructs A137Bpa and A254Azi is less than in HEK cells; E_{rev} from -52mV to -45 mV, is it significant? It thus looks like that the current properties (amplitude and E_{rev}) differ according to the expression system (linked to endogenous Orai2-Orai3 or STIM1-STIM2 expression levels?). Please comment.

10. Fig 8 g-i: It would have been more convincing to show NFAT translocation (or any downstream signals) on cells different from HEK, to persuade the reader that indeed this new tool could be valuable to study signaling pathways downstream of Orai1-induced Ca²⁺ entry.

11. Orai1 and STIM1 KO cells were used, but what about possible regulation by STIM2? Did the authors try on DKO cells (STIM1-STIM2 KO)?

12. In the same line, Orai1 forms heteromers (e.g. Zhang et al 2020 Cell Calcium) did the authors try the TKO cell also? How do you know that all current comes from your modified channel and not from a heterocomplex where one channel activation leads to activation of the endogenous? This could be an interesting biological question to ask with this system.

Minor comments:

1. How strong is the UV pulse used in fig 1 and S1? mW? please specify the mW per cm² (power source)

2. As well on Fig S3, what does the % of UV light refer to? You should put there the Joules or mW/cm² in order to know the real photonic force applied.

3. It is not always clear whether the experiments were performed in cells overexpressing STIM1 together with Orai1. In the material and methods, it is written as if all experiments were done with both Orai1 and STIM1 transfection, which is obviously not the case and thus it raised questions for couple of figures (ex fig 2, 6, 8).

4. How long does the UAA-channel activation by UV illumination lasts? Can Ca²⁺ entry triggered by UV illumination leads to Ca²⁺ overload of the ER?

5. A137Bpa presents an important basal Ca²⁺ entry, is the channel expression toxic to cells?

6. Fig S8a: When was the UV light applied? Before or after store depletion? That should be stated. In Fig S8b, was STIM1 overexpressed?

7. Fig 8i: the scale bars are missing.

8. It would be worth to determine the sensitivity of the 3 UAA mutants to other well-known Orai1 channel blockers, such as GSK 7975A or Synta66 (for ex).

Reviewer #1 (Remarks to the Author):

In this study Maltan et al reports successful incorporation of the photoreactive unnatural amino acids (UAA), p-benzoyl-L-phenylalanine (Bpa) and p-azido-L-phenylalanine (Azi), at three strategic positions A137, L174 and A254 located in the second, third and fourth transmembrane regions of the Orai1 channel, respectively. They show that illumination of UV light onto cells expressing either one of the engineered channels induces a strong calcium flux independently of ER calcium depletion or co-expression of STIM1. Out of the three UV activated currents, the one induced by A137Bpa exhibits several of the key characteristics of the STIM1 activated Orai1 current suggesting that a similar active conformation of Orai1 is stabilized under both conditions. Further in line with a significant similarity between the STIM1 and UV induced Orai1 currents, UV-induced Ca²⁺ flux in Jurkat or RBL cells activates downstream signaling through the calcineurin-NFAT pathway. Hence, the authors generated a novel method for UV induced activation of Orai1 channels in a manner that appear to closely mimic the native activation mechanism by STIM1.

Overall, the study represent a major technological breakthrough, the experiments are well planned and executed and the main conclusions of the work are (for the most part) supported by wealth of new data. My assessment is that this is an important study that will likely be of interest to a broad scientific community. There are only a few points that should be addressed prior to publication of the manuscript.

We thank the reviewer for the constructive comments.

1. The analysis of FCDI by the different engineered Orai1 channels is lacking. This type of CDI is usually characterized by two (fast and slower) decay constants and is sensitive to mutations in either STIM1 or Orai1. The traces shown in figure 7A suggest that while the overall magnitude of inactivation by the native (STIM1-Orai1) and artificial (Orai1A137Bpa) is similar the kinetics of inactivation may be different. Please include analysis of the respective tau's for each of the Orai1 mutants. Also, it is not clear that the decay in Orai1 A137Bpa (or in the other mutants) current represent CDI as the control experiment using high levels of BAPTA for fast chelating of incoming Ca²⁺ is missing. Please provide this control for Orai1 A137Bpa. Finally, there is no explanation for how FCDI is maintained in Orai1 A137Bpa without STIM1. What enables the Ca²⁺ sensitivity of the process? Please discuss these issues.

We analyzed the FCDI of the different light-sensitive Orai1 mutants and provide the fast and slow decay constants in Table 1.

In addition, we performed FCDI experiments using a hyperpolarizing potential to -70mV for Orai1 A137Bpa in the presence of BAPTA in the pipette. Interestingly, in contrast to a comparable extent of FCDI of STIM1+Orai1 and Orai1 A137Bpa currents in the presence of EGTA in the pipette (Figure 7c, Figure S11c,d), in the presence of BAPTA, we detected an increased FCDI for Orai1 A137Bpa compared with STIM1+Orai1 currents (Figure 7e, Figure S11c,d). Or in other words: FCDI of STIM1+Orai1 currents was more pronounced in the presence of EGTA than in the presence of BAPTA as expected, but Orai1 A137Bpa currents showed intriguingly a comparable (or even reduced (at 1500ms)) extent of inactivation in the presence of EGTA compared to BAPTA (Figure 7d, Figure S11c,d).

To follow up the differences in the extent of inactivation using EGTA and BAPTA of wild-type compared with Orai1 A137Bpa currents, we also recorded FCDI at lower hyperpolarizing potentials (-110mV and -90mV). As expected, clear inactivation is seen for STIM1+Orai1 currents using EGTA, which occurs to a reduced extent using BAPTA. In contrast, UV-activated Orai1 A137Bpa currents show higher or comparable inactivation using BAPTA compared to EGTA (Figure S11a-d). Using instead of Ca²⁺, DVF solution at the extracellular side significantly reduced the extent of inactivation in all cases (Figure 7f,

Figure S11c-d). This clearly indicates a Ca^{2+} -dependent FCDI of Orai1 A137Bpa, but with altered extents of inactivation using different Ca^{2+} buffering.

A similar behaviour has been recently published¹ for the mutation of T92 (T92W) in TM1, a position located in close proximity of 6Å to A137 in TM2. Yeung et al.¹ explained the distinct pattern of FCDI by an enhanced Ca^{2+} sensitivity of the Orai1 mutant compared to Orai1 wild-type, leading to higher FCDI upon global and local Ca^{2+} buffering (BAPTA). In contrast, when only global Ca^{2+} was buffered (EGTA), they suggested that a steady-state submembrane Ca^{2+} level develops at the holding potential (+30mV), thus reducing or preventing inactivation upon hyperpolarizing steps¹.

Unique FCDI of Orai1 A137Bpa was reset by STIM1, at least in part, to that of STIM1-mediated Orai1 currents (Figure S11e-f), as also observed for Orai1 T92W¹.

Overall, these results suggest that FCDI is an intrinsic property of Orai channels that is further tuned by STIM1 coupling to Orai1 C-terminus, but possibly also to other cytosolic segments. Our findings open new avenues for characterizing the key determinants mediating inactivation and Ca^{2+} -sensing of Orai1.

An interpretation of these results together with the role of STIM1 in inactivation is now included in the discussion section (p.12-13)

2. There are interesting cell type specific effects regarding current magnitude and inactivation of Orai1 A137Bpa and A254Azi. For example, UV induced currents mediated by Orai1 A254Azi show a prominent slow current inactivation in HEK293 cells, a more moderate one in Jurkat cells while the inactivation is lacking in RBL cells. Also, the relative UV-induced current magnitude of Orai1A137Bpa and Orai1A254Azi is different in the different type of cells. There is no reference or discussion of these effects. Is the inactivation of Orai1 A254Azi calcium dependent? Why is inactivation lacking only in RBL cells? What may lead to these differences in current densities or inactivation of the two engineered channels in the different type of cells? These issues deserve more attention since in addition to STIM1 there are several other proteins like CRACR2A, SARAF or STIMATE that could affect Orai1 function.

We carefully investigated Orai1 A137Bpa and Orai1 A254Azi currents in RBL-2H3 cells and detected reduced currents of the respective mutants in this cell type, which is likely attributed to lower expression levels (Figure S6e). Indeed, an increase in UV-mediated Ca^{2+} currents correlated with enhanced expression levels in HEK293 cells (Figure R1).

Figure R1: Correlation of Orai1 fluorescence intensity with current density. The higher the expression level, the higher is the current density.

Furthermore, we examined the slow inactivation of Orai1 A254Azi in more detail and discovered that it was Ca^{2+} dependent. Enhanced Ca^{2+} permeation obtained by either increasing UV intensities or increasing extracellular Ca^{2+} concentrations led to pronounced inactivation (Figure S4c). Interestingly, in the absence of Ca^{2+} tiny currents remain

indicating also Na⁺ permeability. Also inactivation remained to a certain extent, suggesting that either Na⁺ is still triggering inactivation or inactivation is further defined by a structural relaxation process after photocrosslinking to an energetically favorable conformation. These observations will be characterized in depth in future studies.

Moreover, the extent of slow inactivation was much more pronounced in HEK293 cells compared to RBL-2H3 and Jurkat T cells. The extent of slow inactivation is likely linked to lower expression levels as shown exemplarily in RBL-2H3 cells (Figure S6e). Nevertheless, we cannot exclude that the formation of heteromeric complexes or accessory proteins affect the unique slow Ca²⁺-dependent inactivation of Orai1 A254Azi.

Cell-type specific effects and inactivation of Orai1 A254Azi are now discussed on p. 12, 1st & 2nd paragraph.

3. In Figure 5a – results indicate no change in current magnitude upon UV illumination of the double mutants harboring the V102A mutation. Since STIM1 binding to this Orai1 mutant alters ion selectivity it is important to know whether UV illumination exerts the same effect.

We investigated the effect of V102A on the reversal potential of UV-activatable mutants. All double mutants (Orai1 V102A A137Bpa,) exhibited a V_{rev} higher than Orai1 V102A even before UV light, which was almost but not fully in the range of STIM1-Orai1 wild-type currents (Figure S8b-h). This may be due to an additional effect of the UAA substitution on the pore. V_{rev} did not change after UV activation and/or STIM1 coupling (Figure S8b-h). However, because the currents of the double mutants do not or only marginally change upon UV light irradiation, it remains unclear whether photocrosslinking occurs. It is possible that in these double mutants, although they are constitutively active like Orai1 V102A, the conformation is altered such that no photocrosslinking can occur and thus the reversal potential remains unchanged. Indeed, we detected a complete loss of function in Orai1 H134A after insertion of A137Bpa, L174Bpa, or A254Azi even in the presence of STIM1, suggesting that the combination of these mutations leads to a conformational change that prevents any type of activation.

4. The manuscript will benefit from additional editing to correct for awkward phrasings, grammar issues and typos (including reference duplication). More specifically, please work harder to tighten the discussion section, omit sections in which results are simply restated and replace them with new ones that address how and to what purpose the UV-induced Orai1 activation presented in this study may be used in future studies.

We carefully checked the text, tightened the discussion section and added new sections, as indicated by the red-labeled sections.

References:

1. Yeung, P. S.-W. , Yamashita, M. & Prakriya, M. A human tubular aggregate myopathy mutation unmasks STIM1-independent rapid inactivation of Orai1 channels. Biorxiv (2022)

Reviewer #2 (Remarks to the Author):

Maltan et al, describe how a set of UV-photocrosslinking unnatural amino acids (UAA) incorporated at different positions of the TM domains of ORAI1 channel can gate the channel in a manner resembling physiological stimulation by STIM1. Authors screen different mutants with two types of modifications, p-benzoyl-L-phenylalanine (Bpa) and p-azido-L-phenylalanine (Azi), ending up with 3 mutants, each in a different TM (TM2, 3 and 4) that allow Orai1 opening upon UV exposition. Two of the mutants (A137Bpa and S254Azi) also display small basal Ca²⁺ entry, similar to Orai1 WT. Authors validate channel activity with Ca²⁺ imaging using the R-Geco2.1 probe, and whole-cell patch-clamp recordings, the two major techniques used in this work. Experiments are well done, and data are convincing, however despite all the hard work and efforts that one can appreciate, I do not see a relevant biological question, neither asked nor answered in the paper. For example, the authors characterize and describe their “tool” comparing it to STIM1 endogenous gating, Orai1 GoF and LoF mutations and N-terminal truncations to end up exploring NFAT translocation as a proxy of signaling. To my understanding they fail to go beyond, after validation of the tool, to describe for example a novel sequential step on the intrinsic Orai1 activation rearrangement, or to convince that the UV-photocrosslinking Orai1 could be a valuable tool to study signaling pathways in physiological context.

We thank the reviewer for the constructive comments.

We agree with the reviewer that we introduced in particular a novel tool in the CRAC channel machinery and provide extensive proof for its functionality. Nevertheless, we also provide novel starting points for future investigations. Generally, our work provides the basis for novel insights in the CRAC channel machinery as stated in the discussion: “... the use of light-sensitive UAAs opens up a new dimension allowing remote real-time and dynamic monitoring of CRAC channel structure and function at the amino acid level, which overcomes the limited resolution of traditional technologies.” (p. 11, l. 36). Specifically, our initial screen revealed that “the sites which allowed strong UV light-induced activation are located close to and within the cytosolic helical extension of the TM domains” (p. 11, l. 32). This already stimulated our ongoing research and we meanwhile identified via a more detailed screen comprising all TM domain positions, that in particular TM3 contains most positions which enable upon UAA incorporation the transfer of light-sensitivity. We identified that most of these critical positions are all located at the TM3-TM4 interface, which allowed us to resolve inter-TM domain motions critical for Orai1 pore opening. The wealth of information we generated in this regard by a combined approach of functional and computational methods together with genetic code expansion are currently put together in a separate manuscript.

Moreover, the most critical locus A137 opens up novel directions to study the interplay of TM2 with the basic region of the pore. We currently assume that A137Bpa crosslinks with a residue in TM1 to allow widening of the pore and Ca²⁺ permeation. Moreover, this communication seems to be critical for FCDI and Ca²⁺ sensing of Orai1, as outlined in detail in point 8 and the results and discussion section of the manuscript (p12-13). Also, in this aspect we have ongoing studies running.

Overall, the library of light-sensitive Orai1 mutants will be valuable to dissect structural features determining biophysical properties along with still unknown binding interfaces of CRAC channels, which will stimulate structural and computational studies. In vivo, this currently emerging technology holds the potential to provide fundamental insights into the biology of native ion channels and uncover new targets beneficial for human therapy (p. 13, last paragraph of discussion).

More specifically:

Figures 1 and S1

1. On Fig 1, the small check marks refer to “ no significant change in activity of Orai1 mutants compared to Orai1 wild-type before UV light exposure, but above threshold line”. From Fig S1, both A254Azi and A137Bpa mutants, but specially A137Bpa have a strong basal Ca²⁺ entry before UV illumination. That actually is in line with the traces shown of Fig 2. Thus, I don't understand the criteria to classify large or small basal activity of UAA mutations before UV, for me A137Bpa has a strong basal Ca²⁺ entry (more than 1.3 intensity of R-Geco1.2, not at all comparable to Orai1 WT). The criteria must be understandable. Furthermore, what is the threshold line of Fig S1? The asterisks for the significance are not clear. Significant compared to what? What the different asterisk colors refer to? P values should be provided.

We apologize for the misleading labelling and revised the table in Figure 1. As we discovered no significant difference between Ca²⁺ entry of Orai1 wild-type compared to Orai1 A137Bpa/A254Azi expressing cells before UV light irradiation, we originally decided to put a small checkmark there. Now we tested the individual mutants showing constitutive Ca²⁺ entry for significant differences in Ca²⁺ levels reached in 0mM Ca²⁺ compared to 2mM Ca²⁺. In case of a significant difference, we put a large checkmark in the table, as now performed for Orai1 A137Bpa and Orai1 A254Azi.

Because we detected very small but significant changes in Ca²⁺ influx before and after UV light irradiation for some mutants (e.g. P245Bpa, M243Bpa, V191Bpa), we defined a threshold line at a normalized intensity of 1.1 that depicts naturally occurring variations in Ca²⁺ influx in Orai1-expressing wild-type cells. Only an increase in Ca²⁺ influx above this threshold line was defined as a significant increase.

We clearly indicated now in the legends that the black asterisks show the significance between Ca²⁺ levels observed in 0mM versus 2mM Ca²⁺-containing solutions before UV light. The colored asterisk exhibits the significance between Ca²⁺ entry before versus after UV light for the respective mutant. F and p values for all combinations are provided in the additional supplementary table 1 (Excel file).

2. Figure 2: Orai1 A137Bpa and S254Azi expressing cells display a basal Ca²⁺ entry after switching to 2 mM Ca²⁺ (panels a & c). However, a basal current is not visible before UV illumination on whole-cell recordings for the same UAA modified channels (panels d & f), while the recordings are in 10 mM Ca²⁺. How to interpret this lack of basal current? Did the “leak” disappear because of current subtraction? Is the basal Ca²⁺ entry also observed in absence of STIM1 (STIM1-Orai1 DKO cells)?

We carefully checked our data and performed additional experiments. Interestingly, Ca²⁺ imaging (50% constitutively active) is more sensitive to small constitutive activity than electrophysiology (~5% constitutively active). The reason for low constitutive activity observed in patch-clamp experiments seems to underlie the fact that mostly only moderately transfected cells could be tested. Indeed, highly transfected cells exhibited constitutive activity, however, could be rarely detected in electrophysiology (Figure S4a,b), either due to low seal stability or due to slightly distinct treatment of cells for Ca²⁺ imaging versus electrophysiology (cells for electrophysiology are reseeded 6-10 hours before the experiment; see methods p.15, l.19). Of course, all currents were corrected by traces obtained after La³⁺ perfusion. Also, in STIM1/Orai1 DKO cells we discovered in Ca²⁺ imaging experiments constitutive activity, which was less pronounced in electrophysiology experiments, likely due to lower transfection efficiency of successfully tested cells.

3. Why the 10% UV light recordings on Fig. S3 panels abc have such a reduced 2mM calcium

response before UV? This should be comparable to all conditions, right? It is difficult to conclude whether the cells had a decreased intrinsic activity that could hardly be further enhanced with UV, or whether the small response to 10% UV really reflect the response to a reduced stimulation?

We apologize for the misunderstanding. We did not apply different intensities of UV light, but recorded R-GECO1.2 using different intensities in Figure S3a-c. With 10% (0,1 mW/cm²) light intensity for R-GECO1.2 detection, we detect low Ca²⁺ levels.

Using different UV intensities (at 0,7mW/cm² or higher) we see similar basal Ca²⁺ entry (Figure S3d).

4. p.5 last sentence: "This indicates efficient UAA incorporation within Orai1 A137Bpa-hexamers" derived from S5, is totally speculative, authors cannot conclude the stoichiometry of the channel based on their analysis. Could the authors discuss how they reached such a conclusion? Otherwise, I would suggest being more descriptive and less conclusive.

We toned down this speculative conclusion (p.5, l.32).

5. Fig 4 e,f is hard to understand, and the relevance of such experiments is questionable. On panel f, the light green bars should be labelled STIM1 + and UV -, instead of STIM1 - and UV-; correct? In addition, the condition Orai1 A137Bpa without STIM1 after store depletion and 2mM Ca²⁺ readdition seems to induce larger Ca²⁺ entry than without store depletion (Fig 2a). Is that an effect of STIM2?

The major goal of the experiments in Figure 4e,f is to show that the return of STIM1 to the resting state does not affect the activation and maintenance of the active state upon photocrosslinking. We have relabelled the diagram accordingly. Experiments shown in a diagram were performed in pairwise comparison per day. In new experiments, we performed a direct comparison of UV-induced Ca²⁺ entry of Orai1 A137Bpa in the absence of STIM1 with compared to without store-depletion. In both cases, we recorded a comparable maximum Ca²⁺ entry which was not significantly different (Figure 4e,f).

6. Fig 5: what is the Erev of Orai1 UAA V102A? Are they non-selective channels like the ones with V102A alone? According to current amplitude versus Ca²⁺ elevation (Fig. 5a&b) it seems to be the case, in particular for the L174Bpa mutant, as the current is very large compared to Ca²⁺ amplitude. Does the expression of STIM1 modifies Erev when the UAA mutations are present together with V102A?

We investigated the effect of V102A on the reversal potential of UV-activatable mutants. All double mutants (Orai1 V102A A137Bpa,) exhibited a V_{rev} higher than Orai1 V102A even before UV light, which was almost but not fully in the range of STIM1-Orai1 wild-type currents (Figure S8b-h). This may be due to an additional effect of the UAA substitution on the pore. V_{rev} did not change after UV activation and/or STIM1 coupling (Figure S8b-h). However, because the currents of the double mutants do not change upon UV light irradiation, it remains unclear whether photocrosslinking occurs. It is possible that in these double mutants, although they are constitutively active like Orai1 V102A, the conformation is altered such that no photocrosslinking can occur and thus the reversal potential remains unchanged. Indeed, we detected a complete loss of function in Orai1 H134A after insertion of A137Bpa, L174Bpa, or A254Azi even in the presence of STIM1, suggesting that the combination of these mutations leads to a conformational change that prevents any type of activation.

7. p.9 1st paragraph: “ all other GoF mutations in the different TM domains, except the slight constitutively active V181A mutant, drastically interfered either with the general function or at least with photocrosslinking-induced activation”. I disagree, as the P245L mutation did not interfere with UV activation of all 3 UAA mutants, as shown on Fig 5a,b.

P245L did not interfere with the UV activation of Orai1 L174Bpa and Orai1 A254Azi, but still in part with Orai1 A137Bpa (due to lower activity of Orai1 A137Bpa P245L than Orai1 A137Bpa). Hence, we adapted the sentence to: “ all other GoF mutations in the different TM domains, except the slight constitutively active V181A and in part the P245L mutation, drastically interfered either with the general function or at least with photocrosslinking-induced activation” (p.8, l.37).

8. Fig 7: FCDI data are not convincing. Why was the voltage step to -70 mV instead of -100 mV or even -120 mV that better reveal the current inactivation? Panel b shows normalized current after reaching a plateau. But as the current reactivates after the initial 250 ms, what is the plateau referred to? At which time point? It is rather unexpected that FCDI is present in absence of STIM1 for the A137Bpa mutant channel, as an essential part of FCDI is provided by a STIM1 sequence 475-483. This must be discussed.

In addition to the voltage steps to -70mV, we analyzed the FCDI for voltage steps to -90mV and -110mV in particular for Orai1 A137Bpa, as it mimicked FCDI of wild-type currents best, in contrast to the robust reactivation of Orai1 L174Bpa and Orai1 A254Azi.

As expected, the FCDI was even more pronounced at lower voltage steps. Remarkably, while at hyperpolarizing steps to -70mV FCDI of STIM1+Orai1 and Orai1 A137Bpa currents was comparable, at lower voltages (-90mV, -110mV), we observed a more pronounced extent of FCDI for STIM1+Orai1 currents compared to Orai1 A137Bpa currents (Figure 7c,d, Figure S11a-d). Interestingly, the use of BAPTA instead of EGTA in the pipette decreased the FCDI of STIM1+Orai1 currents as expected but intriguingly left the FCDI of Orai1 A137Bpa comparable or even enhanced it compared to that in the presence of EGTA (Figure 7e,f; Figure S11a-d). The use of DVF solution on the extracellular side completely abolished FCDI under all conditions (Figure 7d,f; Figure S11c,d). This clearly indicates that Orai1 A137Bpa exhibits Ca²⁺-dependent inactivation, however, with reversed properties in response to distinct intracellular Ca²⁺ buffering.

A similar behavior has been recently published for the mutation of T92 (T92W) in TM1, a position located in close proximity of 6Å to A137 in TM2. Yeung et al.¹ explained the distinct pattern of FCDI by an enhanced Ca²⁺ sensitivity of the Orai1 mutant compared to Orai1 wild-type, leading to higher FCDI upon global and local Ca²⁺ buffering (BAPTA). In contrast, when only global Ca²⁺ was buffered (EGTA), they suggested that a steady-state submembrane Ca²⁺ level develops at the holding potential (+30mV), thus reducing or preventing inactivation upon hyperpolarizing steps.

Unique FCDI of Orai1 A137Bpa was reset by STIM1, at least in part (for EGTA, but not BAPTA), to that of STIM1-mediated Orai1 currents (Figure S11e-f), as also observed for Orai1 T92W.

Overall, these results suggest that FCDI is an intrinsic property of Orai channels that is further tuned by STIM1 coupling to Orai1 C-terminus, but possibly also to other cytosolic segments. Our findings open new avenues for characterizing the key determinants mediating inactivation and Ca²⁺-sensing of Orai1.

Moreover, we provided the corresponding fast and slow decay constants (Table 1). The plateau reached means the minimum current reached at t = 250ms after the hyperpolarizing

step. An interpretation of these results together with the role of STIM1 in inactivation is now included in the discussion section (p.12-13)

9. Fig 8: on RBL-2H3 cells, the current of A254Azi is much smaller compared to A137Bpa current, which is very different to what was obtained on HEK cells (A254Azi current is twice as large as A137Bpa current). Is that linked to channel expression? On the other side, the Ca²⁺ selectivity of the two constructs A137Bpa and A254Azi is less than in HEK cells; Erev from -52mV to -45 mV, is it significant? It thus looks like that the current properties (amplitude and Erev) differ according to the expression system (linked to endogenous Orai2-Orai3 or STIM1-STIM2 expression levels?). Please comment.

We carefully investigated Orai1 A137Bpa and Orai1 A254Azi currents in RBL-2H3 cells and detected reduced currents of the respective mutants in this cell type. The lower currents are likely linked to lower expression levels (Figure S6e). Moreover, we investigated the reversal potentials in more detail, and we detected no significant difference (Figure R1). Nevertheless, we cannot exclude that endogenous STIM or Orai isoforms have additional effects, as discussed now in the manuscript (p.12, 1st and 2nd paragraph).

Figure R1: V_{rev} of WT, Orai1 A137Bpa and Orai1 A254Azi currents in different mammalian cells. V_{rev} in RBL-2H3 (RBL) and Jurkat T cells (Jurkat) shows slight, but not significant, reduction compared to HEK293 (HEK) cells.

10. Fig 8 g-i: It would have been more convincing to show NFAT translocation (or any downstream signals) on cells different from HEK, to persuade the reader that indeed this new tool could be valuable to study signaling pathways downstream of Orai1-induced Ca²⁺ entry.

We studied NFAT translocation also in RBL-2H3 cells containing Orai1 A137Bpa or Orai1 A254Azi expressed. Indeed, we discovered upon UV-light irradiation NFAT translocation to the nucleus to a comparable extent like under control conditions. These experiments are included in Figure 8 j-l.

11. Orai1 and STIM1 KO cells were used, but what about possible regulation by STIM2? Did the authors try on DKO cells (STIM1-STIM2 KO)?

We overexpressed Orai1 A137Bpa and Orai1 A254Azi in STIM1/STIM2 KO cells and discovered analogous instantaneous UV induced activation (Figure S6c).

12. In the same line, Orai1 forms heteromers (e.g. Zhang et al 2020 Cell Calcium) did the authors

try the TKO cell also? How do you know that all current comes from your modified channel and not from a heterocomplex where one channel activation leads to activation of the endogenous? This could be an interesting biological question to ask with this system.

We overexpressed Orai1 A137Bpa and Orai1 A254Azi in Orai1/Orai2/Orai3 TKO cells and discovered analogous instantaneous UV induced activation (Figure S6c).

We agree with the reviewer that the possibility of heterocomplex formation exists. Indeed, we showed that coexpression of wild-type Orai1 and Orai1 A137Bpa leads to a decrease in UV-induced activation with increasing Orai1:Orai1 A137Bpa ratio. However, we overexpress our channels at a much higher level than endogenous Orai proteins occur. Moreover, we found similar immediate activation of light-sensitive Orai1 mutants in all KO cell types. Overall, these facts tend to argue against the activation of our modified channel being caused by heteromers. Nevertheless, this issue will stimulate our future research to investigate the effects of photocrosslinking in heteromeric complexes.

Minor comments:

1. How strong is the UV pulse used in fig 1 and S1? mW? please specify the mW per cm² (power source)

We provide the values for the applied light power for UV light irradiation and detection of R-GECO1.2 in the methods section and corresponding figures.

2. As well on Fig S3, what does the % of UV light refer to? You should put there the Joules or mW/cm² in order to know the real photonic force applied.

We provide the values for the applied light power for UV light irradiation and detection of R-GECO1.2 in the methods section and corresponding figures.

3. It is not always clear whether the experiments were performed in cells overexpressing STIM1 together with Orai1. In the material and methods, it is written as if all experiments were done with both Orai1 and STIM1 transfection, which is obviously not the case and thus it raised questions for couple of figures (ex fig 2, 6, 8).

We corrected the methods section in this regard (p.14, l.19). Moreover, it is clearly indicated in the figure legends whether STIM1 has been co-expressed.

4. How long does the UAA-channel activation by UV illumination lasts? Can Ca²⁺ entry triggered by UV illumination leads to Ca²⁺ overload of the ER?

UAA-channel activation by UV light lasts at least 400s or even longer if required (depending on seal stability). Interestingly, Orai1 A254Azi exhibits slow inactivation, which occurs in a Ca²⁺ dependent manner, as described in the manuscript (p.l., Figure S4c-g)

To investigate whether Ca²⁺ entry triggered by UV illumination leads to Ca²⁺ overload in the ER, we investigate ER Ca²⁺ levels using LAR-GECO1 in HEK293 cells expressing either Orai1 or Orai1 A137Bpa compared to mock cells. In all cases the ER-Ca²⁺ concentration remained at comparable levels before and after UV light application and subsequent TG-induced store-depletion led to a similar decrease in Ca²⁺ levels (Figure S6f).

5. A137Bpa presents an important basal Ca²⁺ entry, is the channel expression toxic to cells?

Similar to the overexpression of STIM1 and Orai1 or various Orai1-GoF mutants², the cells survived in a comparable manner to Orai1 A137Bpa expressing cells. To avoid Ca²⁺ excess

in the cells, they were maintained in media containing 0,1mM Ca²⁺ after transfection. Moreover, cells expressing Orai1 A137Bpa show healthy shape and plasma membrane localization (Figure 3j, Figure 4c, Figure S2)

6. Fig S8a: When was the UV light applied? Before or after store depletion? That should be stated. In Fig S8b, was STIM1 overexpressed?

In Figure S8a, UV light was applied after store-depletion at t = 265s, as now stated in legend.

In Figure S8i (originally S8b), STIM1 was not overexpressed. Cells just contain the respective Orai1-GoF mutant.

7. Fig 8i: the scale bars are missing.

We added the scale bars to the respective fluorescence images.

8. It would be worth to determine the sensitivity of the 3 UAA mutants to other well-known Orai1 channel blockers, such as GSK 7975A or Synta66 (for ex).

We chose the CRAC channel blocker CM4620 (also known as AUXORA) as it is structurally similar to Synta66, and is undergoing Phase II clinical trials to treat pneumonia in COVID-19 patients. We compared the inhibitory effect of the CRAC channel blocker CM-4620 on the three light-sensitive Orai1 mutants, Orai1 A137Bpa, Orai1 L174Bpa and Orai1 A254Azi (Figure 7g,h). In all cases, we discovered a significant block of the UV-mediated currents upon application of 10µM CM-4620 (Figure 7g,h).

References:

1. Yeung, P. S.-W. ', Yamashita, M. & Prakriya, M. A human tubular aggregate myopathy mutation unmasks STIM1-independent rapid inactivation of Orai1 channels. *Biorxiv* (2022)
2. Tiffner, A. *et al.* CRAC channel opening is determined by a series of Orai1 gating checkpoints in the transmembrane and cytosolic regions. *J Biol Chem* (2020) doi:10.1074/jbc.RA120.015548.

Reviewers' Comments:

Reviewer #1:

Remarks to the Author:

The authors addressed all previous comments adequately.

I only have minor remaining comments regarding references.

1. Mullins et al showed that FCDI depends on structural features proximal to the channel pore in Orai1 (PMID 26809793). Please reference this study in the discussion on FCDI (line 527).
2. The regulatory role of SARAF (line 487) – this effect of SARAF was first shown by Palty et al (PMID: 22464749). Please add the correct reference.

Reviewer #2:

Remarks to the Author:

I would like to thank the authors for their answers, the additional experimental work done and the new analysis of some data sets.